# Barium in seawater

# Dissolved distribution, relationship to silicon, and barite saturation state determined using machine learning

Öykü Z. Mete[1,2,3,4,*], Adam V. Subhas[2], Heather H. Kim[2], Ann G. Dunlea[2], Laura M. Whitmore[5], Alan M. Shiller[6], Melissa Gilbert[6], William D. Leavitt[3,7], and Tristan J. Horner[1,2,*]

[1]NIRVANA Laboratories; [2]Department of Marine Chemistry & Geochemistry; Woods Hole Oceanographic Institution, Woods Hole, MA 02543, USA; [3]Department of Earth Sciences, Dartmouth College, Hanover, NH 03755, USA; [4]Now at: Department of Earth and Planetary Sciences, Harvard University, Cambridge, MA 02138, USA; [5]International Arctic Research Center, University of Alaska Fairbanks, Fairbanks, AK 99775, USA; [6]School of Ocean Science and Engineering, University of Southern Mississippi, Stennis Space Center, MS 39529, USA; [7]Department of Chemistry, Dartmouth College, Hanover, NH 03755, USA

*Correspondence to: omete@fas.harvard.edu or Tristan.Horner@whoi.edu

## Abstract

Barium is widely used as a proxy for dissolved silicon and particulate organic carbon fluxes in seawater. However, these proxy applications are limited by insufficient knowledge of the dissolved distribution of Ba ([Ba]). For example, there is significant spatial variability in the barium–silicon relationship, and ocean chemistry may influence sedimentary Ba preservation. To help address these issues, we developed 4,095 models for predicting [Ba] using Gaussian Progress Regression Machine Learning. These models were trained to predict [Ba] from standard oceanographic observations using GEOTRACES data from the Arctic, Atlantic, Pacific, and Southern Oceans. Trained models were then validated by comparing predictions against withheld [Ba] data from the Indian Ocean. We find that a model trained using depth, temperature, salinity, as well as dissolved dioxygen, phosphate, nitrate, and silicate can accurately predict [Ba] in the Indian Ocean with a mean absolute percentage deviation of 6.0 %. We use this model to simulate [Ba] on a global basis using these same seven predictors in the World Ocean Atlas. The resulting [Ba] distribution constrains the Ba budget of the ocean to $122(\pm7)\times10^{12}$ mol and reveals systematic variability in the barium–silicon relationship. We also calculate the saturation state of seawater with respect to barite. In addition to revealing systematic spatial and vertical variations, our results show that the ocean below 1,000 m is at equilibrium with respect to barite. We describe a number of possible applications for our model output, ranging from use in biogeochemical models to paleoproxy calibration. Our approach demonstrates the utility of machine learning to accurately simulate the distributions of tracers in the sea and provides a framework that could be extended to other trace elements.

## 1. Introduction

Barium (Ba) is a Group II trace metal that is widely applied in studies of modern and ancient marine biogeochemistry, despite lacking a recognized biochemical function (e.g., Horner & Crockford, 2021). These applications of Ba are based on two empirical correlations relating to its dissolved and particulate cycles. The first correlation relates to the dissolved concentration of Ba, hereafter [Ba], which is strongly correlated with that of the algal nutrient silicon (Si; as dissolved silicic acid; Fig. 1; Chan et al., 1977). Unlike [Si], ambient [Ba] concentrations are faithfully recorded by a number of marine carbonates, such as planktonic (e.g., Hönisch et al., 2011) and benthic foraminifera (e.g., Lea & Boyle, 1990), surface- (e.g., Gonneea et al., 2017) and deep-sea corals (e.g., Anagnostou et al., 2011; LaVigne et al., 2011), and mollusks (e.g., Komagoe et al., 2018). Preservation of these signals means that the Ba content of carbonates can be related to the Ba content of seawater and, by extension, that of Si. Accordingly, the Ba–Si proxy has been applied to understand ocean nutrient dynamics on decadal (e.g., Lea et al., 1989) to millennial timescales (e.g., Stewart et al., 2021).

The nutrient-like distribution of dissolved Ba in seawater is thought to be sustained by the second empirical correlation, relating to cycling of particulate Ba. Particulate Ba in seawater occurs mostly in the form of discrete, micron-sized crystals of the mineral barite ($BaSO_4(s)$, barium sulfate; e.g., Dehairs et al., 1980; Stroobants et al., 1991). Pelagic $BaSO_4$ is an ubiquitous component of marine particulate matter (e.g., Light & Norris, 2021) and constitutes the principal removal flux of dissolved Ba from seawater (Paytan & Kastner, 1996). Pelagic $BaSO_4$ is thought to precipitate within ephemeral particle-associated microenvironments that develop during the microbial oxidation of sinking organic matter (e.g., Chow & Goldberg, 1960; Bishop, 1988). The flux of particulate $BaSO_4$ to the seafloor is correlated with the flux of exported organic matter (e.g., Dymond et al., 1992; Eagle et al., 2003; Serno et al., 2014; Hayes et al., 2021). This correlation means that the accumulation rate of sedimentary $BaSO_4$—or its main constituent, Ba—can be used to trace patterns of past organic matter export on timescales ranging from millenia to millions of years (e.g., Bains et al., 2000; Paytan & Griffith, 2007; Schmitz, 1987; Schroeder et al., 1997).

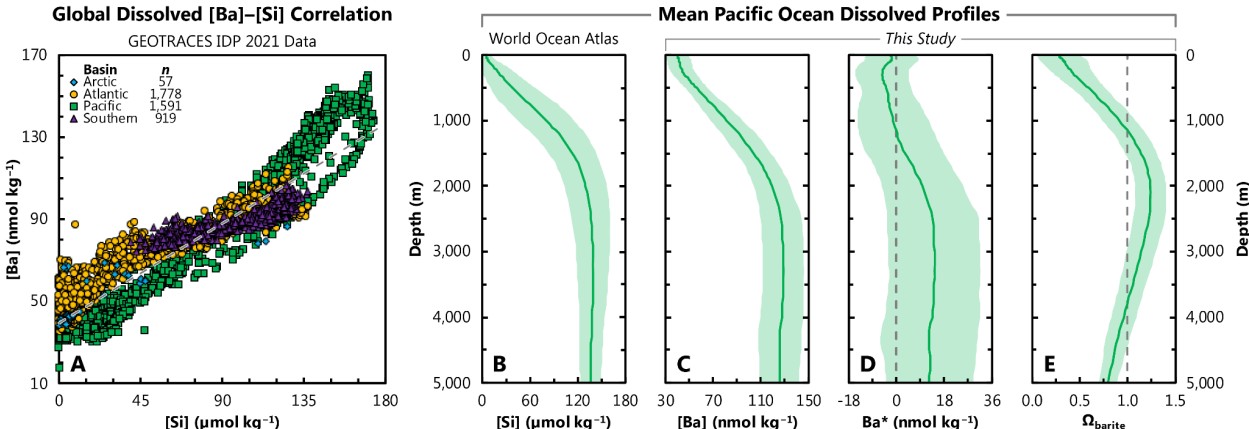

**Figure 1. Distribution of barium in seawater. A.** Property–property plot showing the 4,345 co-located,
core-feature complete dissolved data used in ML model training (Sect. 2). Sample locations shown in Figure
2. Dashed line shows best-fit linear regression through these data, whereby [Ba] = 0.54·[Si] + 39.3. Panels
**B., C., D.,** and **E.** show average Pacific Ocean dissolved depth profiles of [Si], [Ba], Ba*, and $\Omega_{barite}$,
respectively. Solid line denotes the arithmetic mean and the shaded region encompasses one standard
deviation either side of the mean. Dashed line indicates Ba* = 0 (**D**) and $\Omega_{barite}$ = 1 (**E**).
While the Ba-based proxies are valuable, their applications are potentially limited by insufficient
knowledge of the distribution of [Ba]. For example, there is significant vertical and spatial
variability in the Ba–Si relationship (Sect. 3.3.; Fig. 1), which we quantify using Ba* (barium-star;
e.g., Horner et al., 2015):
$$Ba^* = [Ba]_{in\ situ} - [Ba]_{predicted} \qquad \text{[Eq. 1]}$$
where [Ba]$_{predicted}$ is based on the Ba–Si linear regression (Fig. 1):
$$[Ba]_{predicted} = 0.54 \cdot [Si]_{in\ situ} + 39.3 \qquad \text{[Eq. 2]}$$
Here, [Si]$_{in\ situ}$ has units of μmol kg$^{-1}$ and [Ba]$_{predicted}$ nmol kg$^{-1}$; therefore, Ba* also has units of
nmol kg$^{-1}$. The vertical profile of Ba* is rarely conservative (Fig. 1D) and these variations could
introduce uncertainty in the reconstruction of [Si] using Ba.
The relationship between sedimentary BaSO$_4$ accumulation rates and productivity also contains a
significant degree of scatter (e.g., Serno et al., 2014; Hayes et al., 2021). Some of this scatter may
relate to variability in BaSO$_4$ preservation, which is at least partially sensitive to ambient saturation
state, $\Omega_{\text{barite}}$ (e.g., Schenau et al., 2001; Singh et al., 2020; Fig. 1). The saturation state of a parcel
of water with respect to $BaSO_4$ is defined as:
$\qquad \Omega_{\text{barite}} = Q / K_{\text{sp}}$ [Eq. 3]
where $Q$ is the Ba and sulfate ion product and $K_{\text{sp}}$ is the *in situ* $BaSO_4$ solubility product. Discerning
the importance of $\Omega_{\text{barite}}$ on $BaSO_4$ preservation has hitherto been challenging owing to the sparsity
of *in situ* [Ba] measurements. Accurately determining the global distribution of [Ba] would be
valuable for geochemists and oceanographers, and would enable a more thorough investigation of
the effects of preservation on $BaSO_4$ fluxes and refinement of the Ba–Si nutrient proxy.
A powerful way of interrogating oceanic element distributions is through modeling. Broadly, there
are two modeling approaches relevant for simulating [Ba]: mechanistic (i.e., theory driven) and
statistical modeling (i.e., data driven; e.g., Glover et al., 2011). In mechanistic or process-based
modeling, model outputs are derived from sets of underlying equations that are based on
fundamental theory. As such, mechanistic model outputs can be interrogated to obtain
understanding of processes and their sensitivities. However, creating a mechanistic model of the
marine Ba cycle requires embedding a biogeochemical model of $BaSO_4$ cycling within a
computationally expensive global circulation model. Although the computational cost associated
with building mechanistic models has been reduced by the development of ocean circulation
inverse models (e.g., DeVries, 2014; John et al., 2020), this approach still requires detailed
parametrizations of the marine Ba cycle, which do not currently exist. In contrast, statistical models
are based on extracting patterns from existing data and using those relationships to make
predictions. Statistical models encompass a wide variety of approaches ranging from regression
analysis to machine learning (ML). Of particular interest to our study are ML models, which can
make predictions without any explicit parameterizations of causal relationships. Machine learning
models are computationally efficient and can be highly accurate, though they offer limited
interpretability. Machine learning is increasingly being used to solve problems in Earth and
environmental sciences, including simulating the dissolved distribution of tracers in the sea (e.g.,
for cadmium, Roshan & DeVries, 2021; copper, Roshan et al., 2020; iodine, Sherwen et al. 2019;
nitrogen isotopes of nitrate, Rafter et al., 2019; and zinc, Roshan et al., 2018).
The goal of this study is to obtain an accurate simulation of [Ba], which ML makes possible even
in the absence of a process-level understanding of the marine Ba cycle. We tested thousands of
ML models that were trained using quality-controlled GEOTRACES data from the Arctic,
Atlantic, Pacific, and Southern Oceans, supplemented by Argo, satellite chlorophyll, and
bathymetry data products (Sect. 2.). Models were tested for their accuracy by simulating [Ba] in
the Indian Ocean and comparing predictions against observations made between 1977–2013. Since
no Indian Ocean data were seen by any of the models during training, we are able to identify
models with high generalization performance (Sect. 2.). We then identify an optimal set of
predictor variables, calculate model uncertainties, and simulate [Ba], Ba*, and $\Omega_{barite}$ on a global
basis (Sect. 5.). This result will be valuable for researchers interested in marine Ba cycling, and
demonstrates the utility of ML to tackle problems in marine biogeochemistry.
## 2. Training and testing data
Machine learning algorithms are adept at making accurate predictions of a target variable by
identifying relationships between variables within large data sets. However, making accurate
predictions first requires that a ML algorithm is trained on existing observations of that variable
alongside a number of other parameters. These other parameters, hereafter termed features, are an
important part of model training; features should encode information that may help the ML
algorithm predict [Ba], otherwise their inclusion may diminish model performance. Features
should also be well characterized in the global ocean, which allows ML models to make predictions
in regions beyond the initial training dataset. We selected 12 model features by considering the
tradeoff between feature availability and presumed predictive power (Table 1). While testing more
features may have resulted in a more accurate final model, we found that many observations of
[Ba] did not have corresponding data for multiple features; thus, including more features would
have meant fewer training data. Moreover, we find that including more than nine features can
actually diminish model performance. As such, we did not evaluate the predictive power of other
features beyond the 12 initially selected.
**Table 1. List of oceanographic parameters selected as model features.** The features tested were
selected based on their presumed predictive power and geospatial coverage.

| # | Parameter Name | Abbreviation | Units | Coverage* |
|---|---|---|---|---|
| 1 | Latitude | Lat. | degrees north (°N) | – |
| 2 | Longitude | Long. | degrees east (°E) | – |
| 3 | Sample collection depth | $z$ | meters (m) | – |
| 4 | Temperature | $T$ | degrees Celsius (°C) | 97.44% |
| 5 | Salinity | $S$ | unitless, but often written in 'units' of PSU or PSS | 97.44% |
| 6 | Dissolved oxygen | $[O_2]$ | $\mu mol\ kg^{-1}$ | 97.44% |
| 7 | Dissolved nitrate | $[NO_3$ | $\mu mol\ kg^{-1}$ | 97.44% |
| 8 | Dissolved phosphate | $[PO_4]$ | $\mu mol\ kg^{-1}$ | 97.44% |
| 9 | Dissolved silicon (as silicic acid) | $[Si]$ | $\mu mol\ kg^{-1}$ | 97.44% |
| 10 | Maximum monthly mean mixed-layer depth | MLD | meters (m) | 88.20% |
| 11 | Mean average annual surface chlorophyll | Chl. $a$ | mg m$^{-3}$ | 93.95% |
| 12 | Bathymetry | Bathy. | meters (m) | 100% |

*Coverage values represent the percentage of data points within the World Ocean Atlas 2018 grid that have available data for a given parameter. Latitude, longitude, and depth have 100 % coverage as these features define the grid itself.

The 12 features used to predict [Ba] and their associated data sources are summarized in Table 1
and described below. The first three features (latitude, longitude, depth) record geospatial
information that defines the location of an observation in three-dimensional space. To avoid
numerical discontinuities, latitude and longitude were introduced into the model as a
hyperparameter consisting of the cosine and sine of their respective values (in radians). Data for
features 1–3 were included in the sample metadata. Features 4–9 encode physical (temperature,
salinity) and chemical (oxygen, nutrients) information that is routinely measured alongside [Ba].
These data were generally available for the same bottle as the [Ba] measurements; however, when
that was not the case, nutrient data were taken from the corresponding location during a separate
cast, or, in the case of oxygen, from linearly interpolated sensor data. The final three features are
independent of depth, meaning that all samples within a given vertical profile exhibit the same
value for MLD (mixed-layer depth), sea-surface chlorophyll *a*, and bathymetry. Features 10–12
were drawn from several data sources. A climatology of MLD (feature 10) was compiled using
the Argo database (Holte et al., 2017). We selected maximum monthly mean MLD as the feature
of interest, as this appears to be the spatiotemporal scale most relevant for influencing [Ba]
distributions (Bates et al., 2017). Feature 11 represents a blended SeaWiFS and MODIS
climatology of chlorophyll *a* that was obtained from the Copernicus Marine Environment
Monitoring Service (CMEMS, 2021). We calculated the mean annual chlorophyll *a* for each grid
cell in the data product and log transformed the data to reduce parameter weighting (e.g., Rafter et
al., 2019). Data for MLD and chlorophyll *a* were extracted at the location of [Ba] observations
using nearest-neighbor interpolation and their values logged in the master record. Bathymetric
information (feature 12) was extracted from one of two sources. Our preferred source was the
sample metadata, which generally included a value for bathymetry. For samples lacking
bathymetric information, we used nearest-neighbor interpolation to extract a value from the
*ETOPO5* Global Relief Model (National Geophysical Data Center, 1993). Occasionally, the
*ETOPO5*-extracted bathymetry was shallower than the deepest observation of [Ba] in a given
vertical profile. In such cases, the bathymetry logged in the master record was set to 1.01 times the
depth of the deepest observation in that profile.
The [Ba] data from the Indian Ocean were collected from several, primarily pre-GEOTRACES
sources (Table 2). As such, these data were generally incomplete for the 12 features used to train
the ML models. Rather than using a mixture of *in situ* and interpolated data, we decided to
interpolate all Indian Ocean data for parameters 4–12. Data for parameters 4–9 were linearly
interpolated from the nearest vertical profile in the World Ocean Atlas 2018 (WOA; Boyer et al.,
2018; García et al., 2018a; 2018b; Locarnini et al., 2018; Zweng et al., 2018) and values for MLD
and chlorophyll *a* were extracted from the aforementioned data products using nearest-neighbor
interpolation. Bathymetric information was obtained from either the WOA or *ETOPO5*. For the
vast majority of most samples, bathymetry was taken as the arithmetic mean of the maximum
depth of the nearest vertical profile in the WOA and the depth at the standard level below. For
example, if the maximum depth at a station was 950 m, the bathymetry was recorded as 975 m,
which is the mean of levels 46 (950 m) and 47 (1,000 m). For profiles with a maximum depth of
5,500 m—level 102, the lowest in the WOA—bathymetry was recorded as either 5,550 m or the
nearest-neighbor interpolated value from *ETOPO5*, whichever was deeper.
**Table 2. Data sources.** Information regarding the source of [Ba] incorporated into the master record.

| Purpose | Region | Expedition ID | Data source | Data Originators (if unpublished) |
|---|---|---|---|---|
| Model training | South Atlantic (Meridional) | GA02 | GEOTRACES IDP 2017 (Schlitzer et al., 2018) | Jose M. Godoy |
| | North Atlantic (Zonal) | GA03 | Rahman et al., 2022 | |
| | South Atlantic (Zonal) | GA10 | Horner et al., 2015; Bates et al., 2017; Hsieh & Henderson, 2017; Bridgestock et al., 2018 | |
| | Southern Ocean (Meridional) | GIPY04 | GEOTRACES IDP 2017 (Schlitzer et al., 2018) | Frank Dehairs |
| | Southern Ocean (Zonal) | GIPY05 | Hoppema et al., 2010 | |
| | Arctic | GIPY11 | Roeske et al., 2012 | |
| | | GN01 | Whitmore et al., 2022 | |
| | Pacific (Meridional) | GP15 | GEOTRACES IDP 2021 (GEOTRACES IDP Group, 2021) | Laura Whitmore, Melissa Gilbert, Emilie Le Roy, Tristan Horner, Alan Shiller |
| | Subtropical South Pacific (Zonal) | GP16 | Rahman et al., 2022 | |
| Model testing | Indian Ocean | GEOSECS | Craig & Turekian (1980) | |
| | | INDIGO 1 | Jeandel et al. (1996) | |
| | | INDIGO 2 | | |
| | | INDIGO 3 | | |
| | | SR3 | Jacquet et al. (2004) | |
| | | SS259 | Singh et al. (2013) | |


This data ingestion process resulted in a master record containing 5,502 observations of [Ba] that
also contained a corresponding value for all 12 core features (Table 1). The record was then split
into a Pareto partition: the first partition was used for ML model training (4,345 observations, 79
% of data; Fig. 1A) and the second for model testing (1,157 data; 21 %). This partitioning was
determined based on the basin from which the sample was collected; data from the Arctic, Atlantic,
Pacific, and Southern Oceans were used in model training, whereas the 1,157 [Ba] data from the
Indian Ocean were reserved for model testing (Table 2; Fig. 2). This location-based separation of
training and testing data was chosen to minimize overfitting, which can occur when the training–
testing separation is randomly assigned (see Sect. 3.2.).

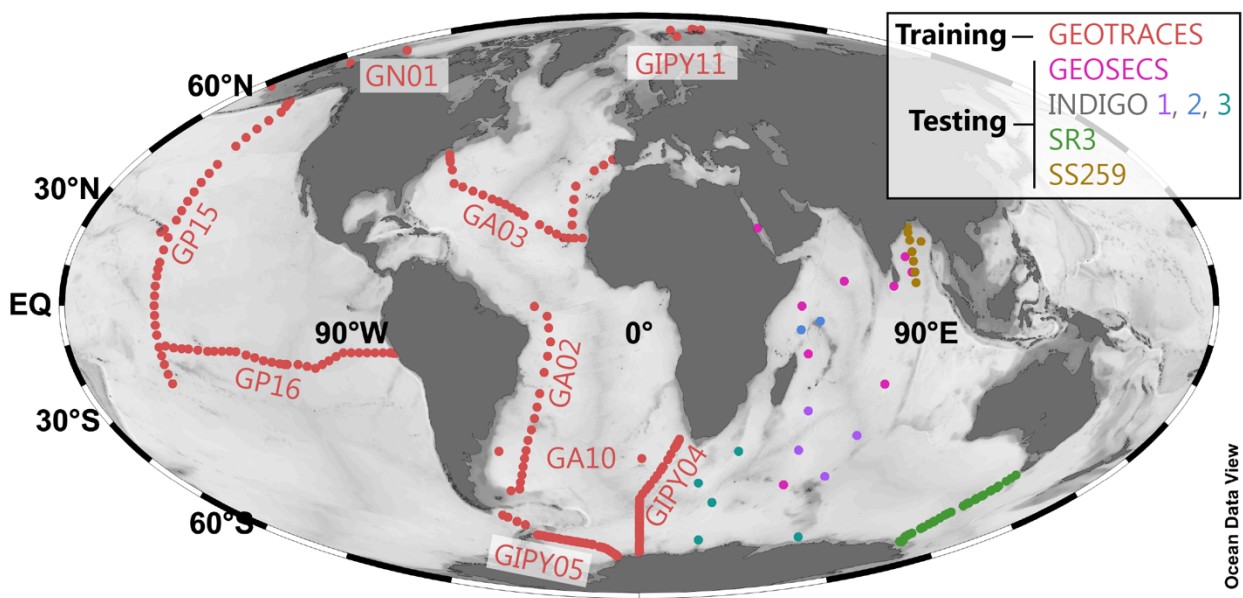

**Figure 2. Geographical distribution of the training and testing data**. The 4,345 core-feature complete
training data (red; Fig. 1) are from the GEOTRACES 2021 Intermediate Data Product (GEOTRACES IDP
Group, 2021); GEOTRACES expedition identifiers are noted next to each section. The $n$ = 1,157 testing
data from the Indian Ocean are color-coded by expedition. Data sources listed in Table 2.

## 3. Methods

In the following subsections we discuss details of the specific ML algorithm that was used for
model development (Sect. 3.1.), explain the model training and testing process (Sect. 3.2.), and
describe how a global prediction of [Ba] was obtained and interrogated (Sect. 3.3.).

### 3.1. Algorithm selection and training

We opted for supervised ML using a Gaussian Process Regression learner, implemented in MATLAB. This particular ML algorithm is non-parametric, kernel-based, and probabilistic, which means that it does not make strong assumptions about the mapping function, can handle nonlinearities, and takes into account the effect of random occurrences when making predictions. Gaussian Process Regression algorithms are widely used in geostatistics, where it is often referred to as 'kriging' (e.g., Cressie, 1993; Rasmussen & Williams, 2006; Glover et al., 2011). This type of algorithm is ideal when working with continuous data that also contains a certain level of noise, such as from measurement uncertainty or oceanographic variation. The MATLAB function, `fitrgp`, was used for model training. A full list of the parameter selections used in `fitrgp` is provided in Table S1. All predictors were normalized and standardized to have a mean of zero and a standard deviation of unity. This process places all parameters on the same relative range and reduces scale dependencies.

A selection of the training data were used to train 4,095 different machine learning models with the goal of finding a model that could accurately simulate the global distribution of [Ba]. The number of models derives from the number of features investigated; each model uses a unique combination of the 12 features in Table 1 and our testing followed a factorial design whereby each feature was either enabled or disabled. This design yields a total of $2^{12}$ unique feature combinations (i.e., levels$^{features}$); however, since it is not possible to train a model with no features enabled, the final number of unique, trainable, ML models with $\geq 1$ features is $2^{12}-1=4,095$. The full experiment list is provided in Section 6. Each of the 4,095 models was trained using the same data and with the same function parameters described in Table S1.

### 3.2. Assessing model performance

Model performance—accuracy and generalizability—was assessed during two phases: training and testing. During model training, the 4,345 observations of [Ba] from the Arctic, Atlantic, Pacific, and Southern Oceans were randomly split into two folds: a training fold containing 80 % of the observations, and a holdout fold containing the other 20 %. Model accuracy was assessed by comparing model-predicted [Ba] against observed [Ba] for the 20 % of the data in the holdout

fold. We then performed additional testing to establish model generalizability. A significant
problem in supervised ML, and particularly Gaussian Process Regression learning, is overfitting:
models may fit the noise in the training data, leading to poor generalization performance
(Rasmussen & Williams, 2006). Since our goal was to develop a global model of [Ba] using
regional training data, we deemed it especially important to identify generalizable models.
Generalizable models were identified through a testing process involving regional cross-
validation; each trained model was used to predict [Ba] for the 1,157 samples from the Indian
Ocean and model predictions were again compared against observations. Importantly, no [Ba] data
from the Indian Ocean were seen by any of the models during training. This process helped to
identify models that may have been overfit to the training data and can further be used to calculate
generalization errors (Sect. 4.1).
The accuracy of trained models was determined by comparing ML model predictions against
withheld data and calculating the mean absolute error (MAE) and mean absolute percentage error
(MAPE), defined as:
$$\text{MAE} = \frac{\sum_{i=1}^{n} \left| [Ba]_{predicted} - [Ba]_{observed} \right|}{n}$$  [Eq. 4]
and:
$$\text{MAPE} = \frac{100\,\%}{n} \sum_{i=1}^{n} \left| \frac{[Ba]_{predicted} - [Ba]_{observed}}{[Ba]_{observed}} \right|$$  [Eq. 5]
respectively, where *n* is the sample size.
Models with lower accuracy exhibit higher errors, whereas models with high accuracy have lower
errors. We calculated MAE and MAPE for every possible feature combination, which enables
quantification of how specific features affect model performance. Likewise, we calculated errors
for each model on predictions made during training (i.e., for the holdout fold) and during model
testing (i.e., during regional cross-validation; Fig. 3). This information is used to quantify
generalization performance; low errors for both training and testing indicate models that are both
accurate and generalizable, whereas models with low training errors and high testing errors might
indicate models that are overfit to the training data.

## 3.3. Global predictions

A select number of models with low MAE and MAPE were used to simulate [Ba] on a global basis. The process by which we selected these models is described in Section 5.1. Global simulations were performed on the same grid as the WOA, which was also used as the data source for features 1–9 (Boyer et al., 2018). The WOA is a 1°×1° resolution data product with around 41,000 stations that contain up to 102 depth levels spanning 0–5,500 m in 5, 25, 50, or 100 m increments. Data for features 10–12 (MLD, chlorophyll *a*, and bathymetry) were also resampled to the WOA grid using the same sources and interpolation methods as described for the Indian Ocean testing data in Section 2. Model outputs were visualized using Ocean Data View software (ODV; Figs. 5–8; Schlitzer, 2023).

A selection of the most accurate models of [Ba] were then used to simulate Ba* and $\Omega_{barite}$. Star tracers, such as Ba*, are valuable for illustrating processes that influence the cycling of elements in the ocean. First defined for N–P decoupling (N*; Gruber & Sarmiento, 1997) star tracers show variations whenever there are differences in the sources and sinks of the two elements being compared. If there are no differences in sources and sinks, the tracer will show conservative behavior because both elements share the same circulation. Barium-star is based on Ba–Si decoupling and was first defined by Horner et al. (2015). The definition of Ba* is shown in Equations 1 and 2. The coefficients in Equation 2 are based on data from the GEOTRACES 2021 Intermediate Data Product and specifically the subset of these data shown in Figure 1. These coefficients differ from previous formulations of Ba* that were based primarily on [Ba] and [Si] data from the Southern and Atlantic Oceans (e.g., Horner et al., 2015; Bates et al., 2017). The global distribution of Ba* was determined by calculating [Ba]*predicted* (Eq. 2) using [Si]*in situ* from the WOA 2018 (García et al., 2018b). The values of [Ba]*in situ* was taken from the ML model output and [Ba]*predicted* was subtracted from this to yield Ba* (Eq. 1).

Values of $\Omega_{barite}$ were computed using the method described by Rushdi et al. (2000), summarized in Equation 3. The numerator, *Q*, represents the *in situ* Ba and sulfate ion product and, in this formulation, depends only on [Ba] and [$SO_4^{2-}$] molality. The denominator, $K_{sp}$, depends on *T*, *S*, and *z* (i.e., pressure) and is calculated in two steps: *in situ T* and *S* are used to calculate the stoichiometric solubility product and then this value is modified by calculating the effect of pressure on partial molal volume and compressibility, which are functions of *T* and *z*. As with the

calculation of Ba*, values of [Ba]$_{in\ situ}$ were obtained from ML models and co-located data for $T$,
$S$, and $z$ were extracted from the WOA (Locarnini et al., 2018; Zweng et al., 2018). Sulfate
concentrations were assumed to be conservative with respect to $S$ using [SO$_4^{2-}$] = 29.26 mmol kg$^-$
$^1$ when salinity = 35 PSU. This latter assumption likely breaks down in certain environments (e.g.,
where [SO$_4^{2-}$] reduction occurs); as such, our model is not used to predict $\Omega_{barite}$ in restricted
basins, such as the Black Sea or Caspian Sea. Given that our estimates of $\Omega_{barite}$ exhibit a MAE of
0.08 (Appendix), we believe that values of $\Omega_{barite}$ between 0.92 and 1.08 are indicative of 'perfect'
saturation with respect to BaSO$_4$.
Output from the most accurate ML models was then used to calculate mean [Ba] and $\Omega_{barite}$ for
each basin, for a series of prescribed depth bins, and for the global ocean. This calculation was
performed by weighting each cell in the model output by its volume, which ensures a fair
comparison between any two points in the model output. We then subdivided the global ocean into
five sub-basins: Arctic, Atlantic, Indian, Pacific, and Southern. Basin boundaries were defined as
per Eakins & Sharman (2010), though we merged the Mediterranean and Baltic Seas into the
Atlantic and considered the South China Sea as part of the Pacific Ocean. Neither [Ba] nor $\Omega_{barite}$
were simulated in the Black or Caspian Seas and thus these regions are not included in the global
mean calculations.
**4. Results**
**4.1. Factors affecting model accuracy**
Here we examine how model performance is influenced by the number and nature of features
included during training. We consider model performance in terms of accuracy and
generalizability, which we quantify using MAE (Eq. 4). We first explore how the number of
features influences model performance (Fig. 3). Here we see that increasing the number of features
generally improves the accuracy of trained models; however, the response differs depending on
whether accuracy is calculated based on comparison to the holdout fold (i.e., during model
training) or to the withheld Indian Ocean data (i.e., during model testing). When considering only
the holdout fold, trained models predict [Ba] with a high level of accuracy—the mean, median,
and most-accurate trained models achieve a MAE of 2.4, 1.7, and 1.3 nmol kg$^{-1}$, respectively.
Similarly, increasing the number of features almost always improves model accuracy; the MAE of

the most accurate model for a given number of features decreases from 6.5 to 1.3 nmol kg$^{-1}$ as the number of features is increased from one to nine, at which point MAE plateaus between 1.4–1.5 nmol kg$^{-1}$ for models with 10–12 features (Fig. 3A).

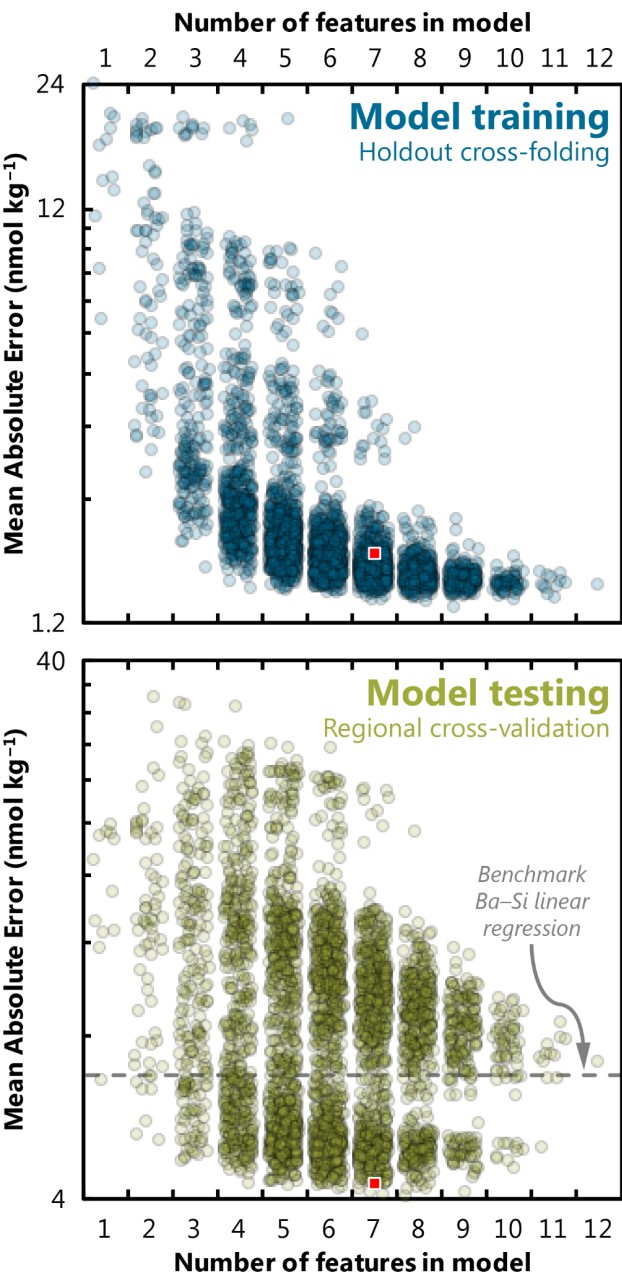

**Figure 3. Effect of feature addition on ML model accuracy.** Accuracy was quantified for each of the 4,095 trained models and quantified here using MAE (note log scale, which differs between panels). The accuracy of trained models is shown for random holdout cross-validation during training (top) and for regional cross-validation during testing (bottom). Square indicates the performance of our favored predictor model, #3080 (see Fig. 4, Sect. 5.1). The accuracy of the Ba–Si linear regression benchmark is shown as a dashed line in the lower panel (MAE = 6.8 nmol kg$^{-1}$). To illustrate data density, points have been randomly positioned within their respective bin and plotted with 80 % transparency.

Moving to the regional cross-validation, the overall performance of models is lower; the same
4,095 trained models achieve a mean, median, and most-accurate MAE for the Indian Ocean
dataset of 8.8, 7.9, and 4.0 nmol kg$^{-1}$, respectively. For comparison, if [Ba] was estimated for these
same 1,157 Indian Ocean samples using the linear [Ba]–[Si] relationship (Fig. 1) and ambient [Si]
as the only predictor, this linear model would achieve a MAE of 6.8 nmol kg$^{-1}$. Thus, there are
1,687 ML models that achieve a superior accuracy to existing methods for estimating [Ba],
offering an improvement of as much as 41 % (Fig. 4). However, regional cross-validation also
shows that the addition of more features may, in fact, degrade model performance. The MAE of
the most accurate model for a given number of features decreases from 6.6 to 4.0 nmol kg$^{-1}$ as the
number of features is increased from one to eight. As the number of features is increased from 9–
12, the MAE of the most-accurate models increases monotonically from 4.1 to 7.1 nmol kg$^{-1}$. The
overall lower performance of trained models during regional cross validation—and the observation
that many of the feature-rich models perform worse than models with fewer features—is indicative
of certain models being over-fit to the training data. Together, these observations suggest that the
optimum number of features needed to accurately predict [Ba] is between six and nine.

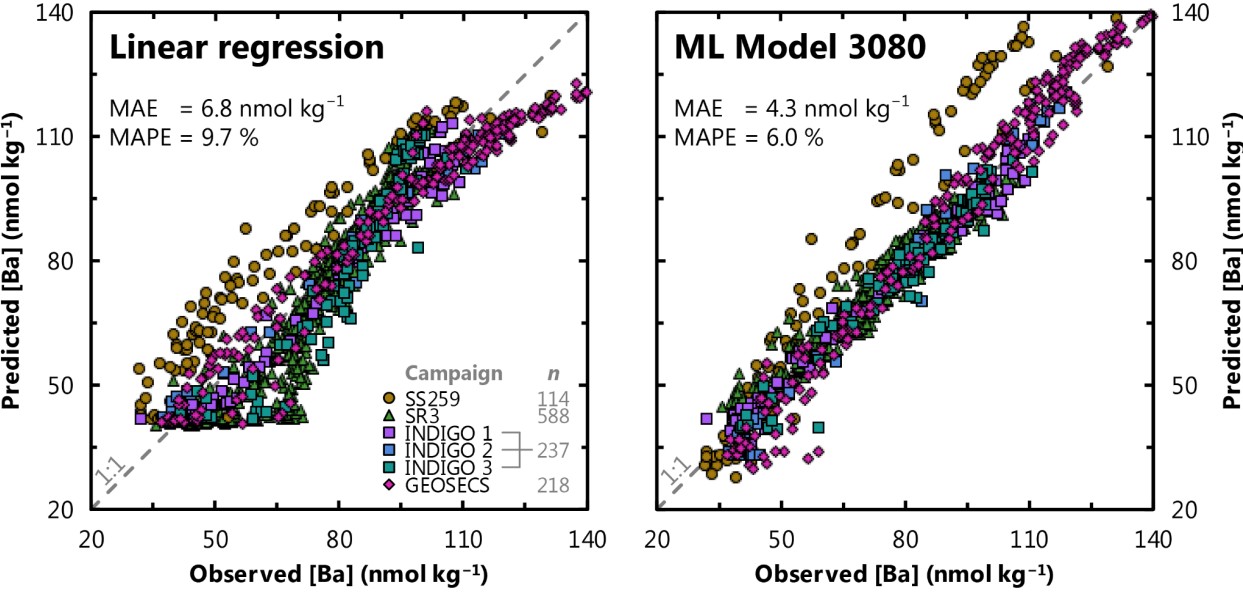

**Figure 4. Comparison of existing and ML methods to estimate [Ba] in seawater.** Left panel shows the
performance benchmark: predicted [Ba] for the Indian Ocean testing data using the [Ba]–[Si] linear
regression and ambient [Si] as the sole predictor. Right panel shows predicted [Ba] using ML model 3080,
which improves on existing methods by more than 37 %. Perfect correspondence between predictions and
observations is indicated b the dashed line marked '1:1.'Data locations and sources are shown in Fig. 2
and Table 2, respectively; *n* refers to the number of testing data for each campaign. Mean Absolute Error
(MAE; Eq. 4) and Mean Absolute Percentage Error (MAPE; Eq. 5) are noted for both models.
We also evaluated the nature of the predictors used to estimate [Ba]. The full factorial experiment
design enables us to perform comparisons between all models that contained a certain feature and
all of those that did not (Sect. 3.1). We quantified the effect of adding a feature by comparing the
absolute and percentage change in MAE relative to the mean MAE of the two sets of models. This
comparison was performed three times: for all 4,095 models based on the holdout cross-folded
training data, for all models using the regionally cross-validated testing data, and again for the
testing data, but only considering those 1,687 models that achieved a superior accuracy compared
to the [Ba]–[Si] linear regression model (Table 3).
**Table 3. Feature addition analysis.** Effect of each feature on model performance for Training and Testing
datasets. Model performance is quantified using MAE, thus all columns have units of nmol kg$^{-1}$ unless
otherwise shown. The Testing analysis is further subdivided into a comparison of all models and 'good'
models, meaning those that achieved superior accuracy than the Ba–Si linear regression (Fig. 1).

| Feature | Training | | | Testing | | | | | | |
|---|---|---|---|---|---|---|---|---|---|---|
| | All models (*n* = 4,095) | | | All models (*n* = 4,095) | | | Good models (*n* = 1,687) | | | |
| | Mean MAE of models with feature | Mean MAE of models without feature | Relative change in MAE | Mean MAE of models with feature | Mean MAE of models without feature | Relative change in MAE | Mean MAE of models with feature | Mean MAE of models without feature | Relative change in MAE | Share of models with feature |
| [Si] | 1.71 | 3.03 | -56% | 7.08 | 10.6 | -39% | 5.06 | 5.50 | -8.3% | 63% |
| z | 1.83 | 2.90 | -45% | 7.94 | 9.70 | -20% | 5.05 | 5.44 | -7.4% | 55% |
| [O$_2$] | 2.03 | 2.71 | -29% | 8.25 | 9.39 | -13% | 5.14 | 5.33 | -3.8% | 54% |
| T | 1.78 | 2.96 | -50% | 7.61 | 10.0 | -27% | 5.17 | 5.31 | -2.8% | 59% |
| [NO$_3$] | 2.09 | 2.65 | -24% | 8.27 | 9.36 | -12% | 5.16 | 5.30 | -2.7% | 53% |
| [PO$_4$] | 2.11 | 2.63 | -22% | 8.24 | 9.40 | -13% | 5.17 | 5.30 | -2.4% | 53% |
| S | 2.02 | 2.72 | -29% | 8.67 | 8.97 | -3.5% | 5.23 | 5.23 | 0.0% | 53% |
| Bathy. | 2.30 | 2.44 | -6.1% | 8.55 | 9.08 | -6.0% | 5.23 | 5.22 | 0.2% | 51% |
| Chl. | 2.25 | 2.48 | -10% | 8.67 | 8.97 | -3.5% | 5.24 | 5.22 | 0.4% | 50% |
| MLD | 2.31 | 2.43 | -4.8% | 8.69 | 8.95 | -3.0% | 5.24 | 5.21 | 0.5% | 50% |
| Lat. | 2.16 | 2.58 | -18% | 8.13 | 9.51 | -16% | 5.32 | 5.11 | 4.0% | 54% |
| Long. | 2.17 | 2.57 | -17% | 11.4 | 6.24 | 58% | 6.45 | 5.19 | 22% | 3% |

This analysis yields three main results. When considering only the holdout cross-folded training
data, the addition of any of the 12 features improves model performance by between –4.8 and –56
%. Excepting longitude, similar across-the-board improvements were observed when considering
only the testing data, though the improvements for most features were more modest (between –3.0
and –39 %). If considering only the 'good' models, six features improved model performance by
–2.4 and –8.3 % ([PO$_4$], [NO$_3$], $T$, [O$_2$], $z$, and [Si]), five degraded model performance by +0.2 to
+22 % (bathy., Chl. a, MLD, lat., and long.), and salinity had no significant effect (Table 3).
Overall, our results indicate that between six and nine features will result in an accurate and
generalizable ML model of [Ba], and that [PO$_4$], [NO$_3$], $T$, [O$_2$], $z$, [Si], and possibly $S$, are likely
to be included as predictors in such a model.

## 4.2. Model outputs

Almost 1,700 models achieved superior accuracy compared to the Ba–Si linear regression
benchmark of 6.8 nmol kg$^{-1}$. We winnow this list to a single model, #3080, in the next section.
We henceforth refer to model #3080 as our favored predictor model, which achieves a MAE of
4.3 nmol kg$^{-1}$ using $z$, $T$, $S$, [O$_2$], [PO$_4$], [NO$_3$], and [Si] as predictors (Fig. 4). Model #3080 is used
to simulate [Ba], Ba*, and $\Omega_{barite}$ on a global basis and to calculate whole-ocean averages. Surface
plots showing the model outputs for the sea surface, 1,000 m, 2,000 m, and 4,000 m are shown in
Figures 5, 6, 7, and 8, respectively.

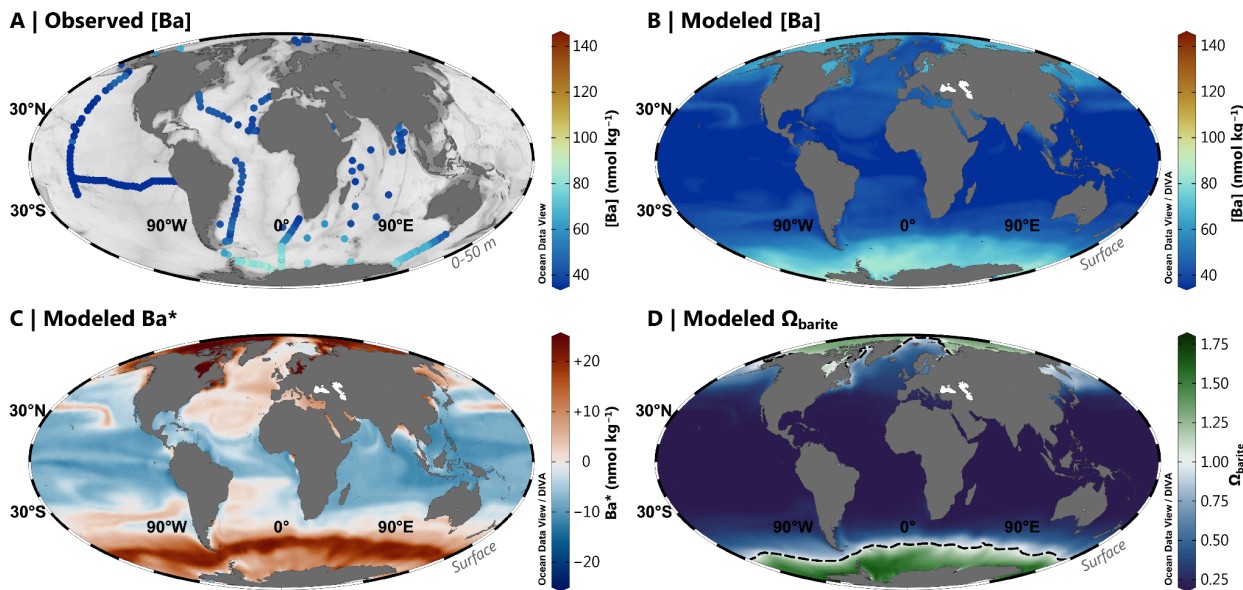

**Figure 5. Barium at the sea surface.** Observed [Ba] between 0–50 m (**A**); Model 3080 [Ba] (**B**), Ba* (**C**),
and $\Omega_{barite}$ (**D**). The dashed line in Panel D indicates the BaSO$_4$ saturation horizon (i.e., $\Omega_{barite}$ = 1.0). Panels
A and B use the *roma* color map, whereas Panels C and D use *vik* and *cork*, respectively (Crameri, 2018).
Color palettes and parameter ranges are the same for the respective panels in Figure 6–8.

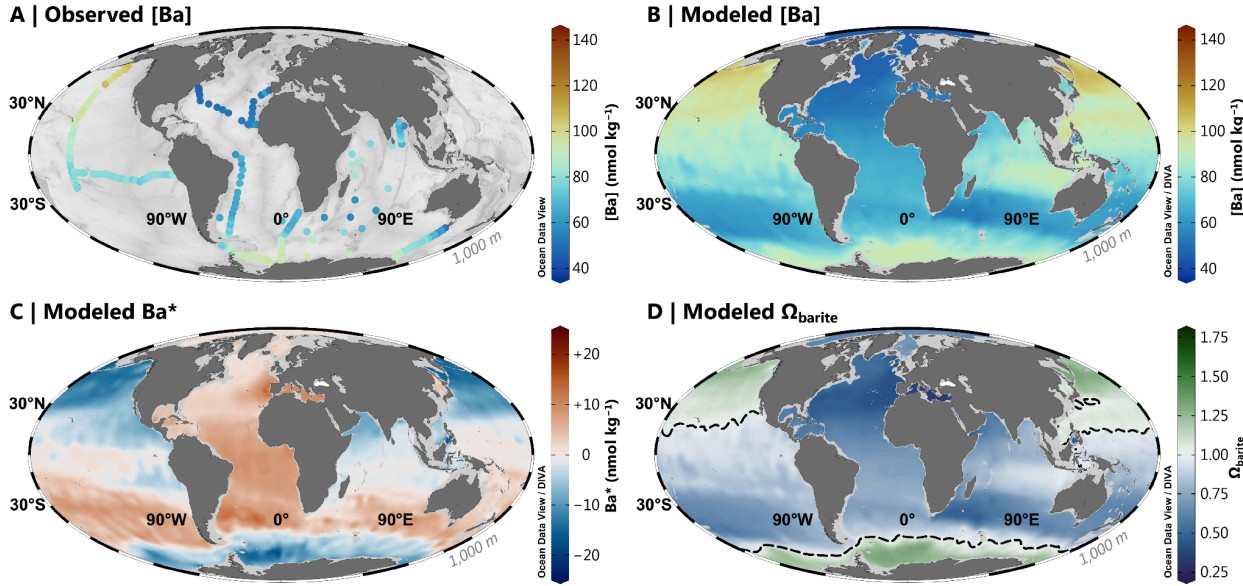

Figure 6. Barium at 1,000 m. Observed [Ba] (A); Model 3080 [Ba] (B), Ba* (C), and $\Omega_{barite}$ (D). The dashed line in Panel D indicates the $BaSO_4$ saturation horizon.

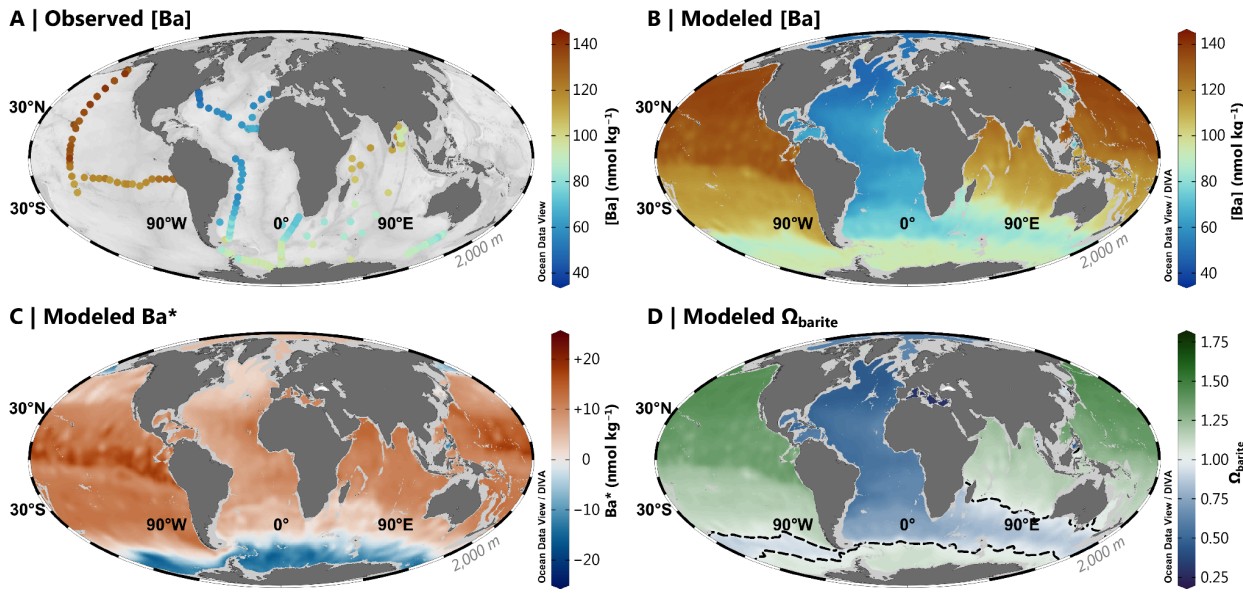

Figure 7. Barium at 2,000 m. Observed [Ba] (A); Model 3080 [Ba] (B), Ba* (C), and $\Omega_{barite}$ (D). The dashed line in Panel D indicates the $BaSO_4$ saturation horizon.

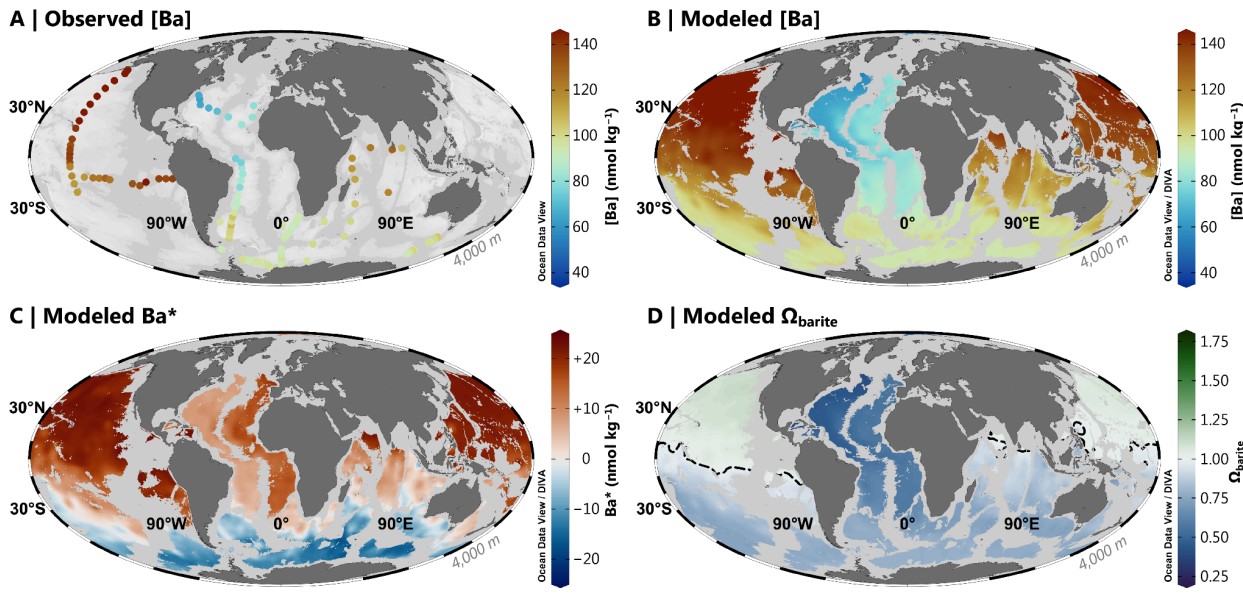

**Figure 7. Barium at 4,000 m.** Observed [Ba] (**A**); Model 3080 [Ba] (**B**), Ba* (**C**), and $\Omega_{barite}$ (**D**). The dashed line in Panel D indicates the $BaSO_4$ saturation horizon.

Model #3080 contains 3,302,570 predictions for each of [Ba], Ba*, and $\Omega_{barite}$ (Sect. 6). Assuming that the MAPE and MAE are good estimates of the prediction error, we estimate that modeled [Ba] and Ba* have uncertainties of 6.0 % and 4.3 nmol kg$^{-1}$, respectively. Uncertainties on $\Omega_{barite}$ were estimated by comparison to literature data, which yields a MAE of 0.08. These estimates are discussed in more detail in Section 5.2 and the Appendix.

Modeled [Ba] ranges from 26.2–156.8 nmol kg$^{-1}$ and the data exhibit an unweighted mean of 72.0 nmol kg$^{-1}$. The range of model #3080 predictions is within the range of [Ba] encountered in the 4,345 training data (17.1–159.8 nmol kg$^{-1}$). This is an important consideration when assessing the accuracy of Gaussian Process Regression models, and we provide additional discussion of this point in the Supplement. Based on our formulation (Eqs. 1, 2), Ba* varies from –27.2 to +27.9 nmol kg$^{-1}$ and possesses an unweighted mean of +2.4 nmol kg$^{-1}$. Values of $\Omega_{barite}$ vary from 0.11 to 1.70 and exhibit an unweighted mean of 0.75. To account for the different volumes represented by each cell in the WOA grid, we constructed a volume-weighted mean of [Ba] and $\Omega_{barite}$ for the ocean as a whole, for each ocean basin, and for a series of prescribed depth bins (Fig. 9). Looking at the ocean as a whole, the probability density function of [Ba] roughly resembles a uniform distribution, with a mean ocean [Ba] of 89 nmol kg$^{-1}$ (Fig. 9A). Within this mean is considerable

spatial and vertical variation. For example, the Arctic Ocean exhibits the lowest volume-weighted
mean [Ba] of 54 nmol kg$^{-1}$, whereas mean Pacific [Ba] = 106 nmol kg$^{-1}$. The Indian Ocean exhibits
a similar mean [Ba] (90 nmol kg$^{-1}$) to the mean of the global ocean. Shallower than 1,000 m, [Ba]
infrequently exceeds 100 nmol kg$^{-1}$, whereas concentrations <45 nmol kg$^{-1}$ are rare below 1,000
m (Fig. 9B).
The probability density function of volume-weighted $\Omega_{barite}$ is more similar to a normal
distribution, albeit with a slight negative skew. Volume-weighted mean oceanic $\Omega_{barite}$ is 0.82. The
Arctic, Atlantic, and Indian Oceans are, on average, undersaturated with respect to BaSO$_4$, all
exhibiting mean $\Omega_{barite}$ ≤0.82. In contrast, the Pacific and Southern Oceans are within uncertainty
of saturation, with mean $\Omega_{barite}$ of 0.97 and 1.04, respectively (Fig. 9C). Values of $\Omega_{barite}$ <0.2 are
mostly restricted to the upper 250 m, whilst values of $\Omega_{barite}$ exceeding 1.5 are exceptionally rare,
found only in the upper 1,000 m of the Southern Ocean. Lastly, $\Omega_{barite}$ tends to increase between
the 0–250 m, 250–1,000 m, and 1,000–2,000 m depth bins, increasing from 0.42, to 0.65, and 0.96,
respectively. Average $\Omega_{barite}$ in the deepest bin (2,000–5,500 m) is slightly lower, with a mean
value of 0.92 (Fig. 9D). Given the accuracy of our model-derived $\Omega_{barite}$ predictions (0.08 to 0.10),
the ocean between 1,000–5,500 m is within uncertainty of BaSO$_4$ equilibrium.

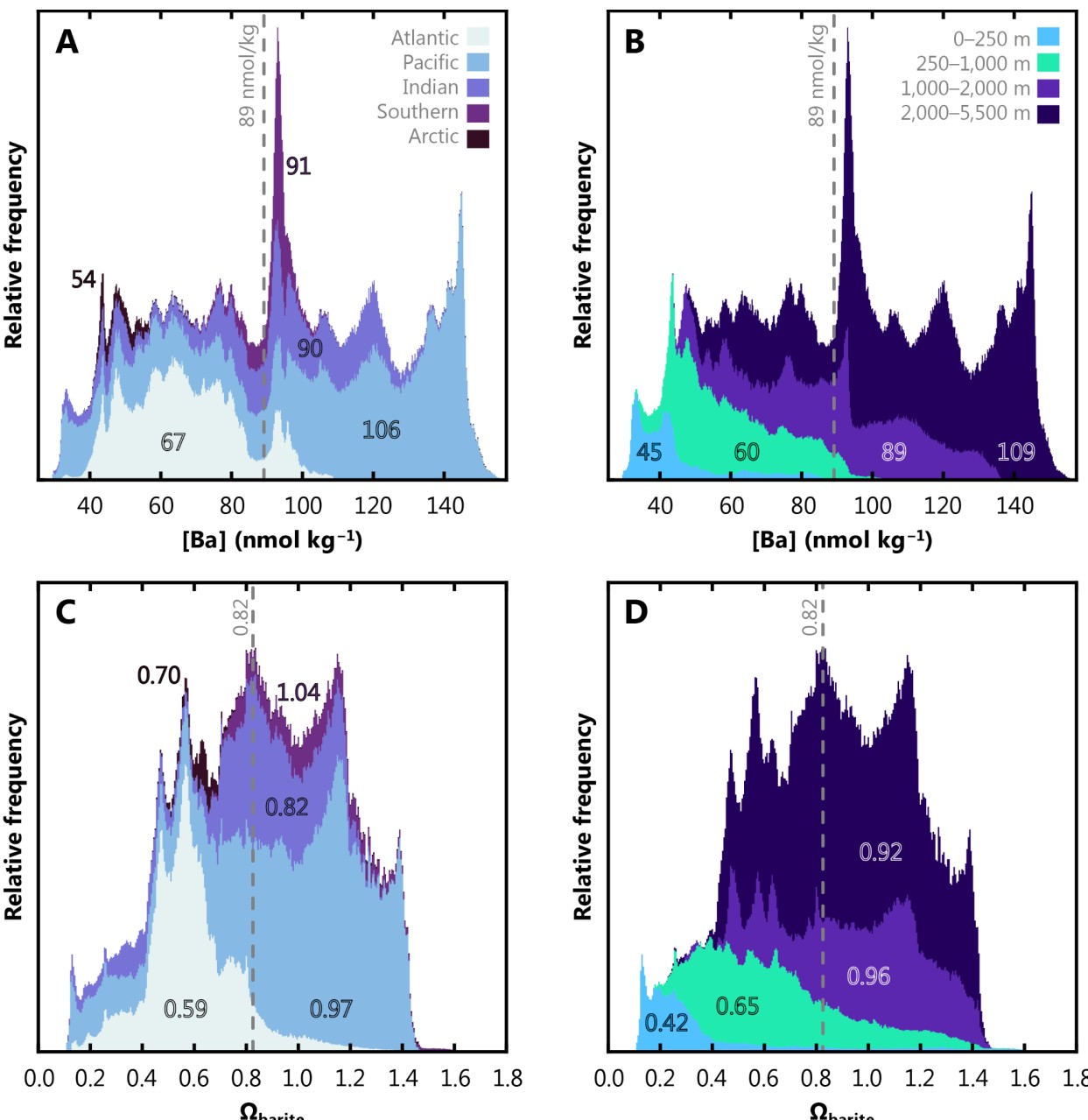

**Figure 9. Stacked, volume-weighted histograms showing the relative frequency distribution of dissolved [Ba] (A, B) and $\Omega_{barite}$ (C, D) in the global ocean.** The left column shows data grouped by basin and the right column shows data grouped by a prescribed depth bin. Numbers in each panel display the mean property value for that bin. Dashed line shows the global mean.

## 5. Discussion

### 5.1. Identification of the optimal predictor model

Choosing a single, optimal model configuration is challenging given the sheer number of skillful ML models. Below we winnow the list from 4,095 to a single model (#3080). We base our winnowing primarily on the results of the regional cross-validation performed in the Indian Ocean, rather than from the errors determined from random holdout cross folding of the training data. We believe that there are three strong reasons for winnowing in this way. First, Gaussian Process Regression Learners tend to fit the noise in the training data, meaning that the training error is significantly lower than the generalization error (Rasmussen & Williams, 2006). Indeed, trained models showed overall lower performance during testing compared to training, which we believe is evidence of overfitting (Fig. 3, Table 3). Second, a generalizable global model should be able to make predictions in regions where it has not already learned anything about the target variable. Our regional cross-validation approach satisfies this consideration since no Indian Ocean data were seen during model training. Third, the Indian Ocean is an ideal basin for testing as it exhibits the full diversity of features expected to influence [Ba] (riverine inputs, oxygen-minimum zones, coastal upwelling, etc.) and constitutes ≈20 % of the global ocean volume. Likewise, the Indian Ocean captures most of the range in [Ba] seen elsewhere in the ocean (Fig. 9); this likely reflects the input of Atlantic waters through the Aughulas leakage, transport of old Pacific waters via the Indonesian Throughflow, and northward spreading of mode and intermediate waters from the Southern Ocean. We thus assume that the Indian Ocean testing errors are a good approximation of the generalization error, which we now use to winnow the list of models.

Our results show that 1,687 of the 4,095 ML models (41 %) produce more accurate predictions of [Ba] than the benchmark Ba–Si linear regression using [Si] as the sole predictor (Fig. 3, Table 3). We focus our winnowing on these 1,687 models as they are superior to existing methods for estimating [Ba] in seawater. Focusing only on these 'good' models reveals significant differences in the information content of the 12 features tested. For example, the inclusion of spatial information in the form of latitude and longitude significantly degrades mean model performance by between +4.0 and +22 %, respectively. While bathymetry, chlorophyll $a$, and mixed-layer depth exhibited only minor influences, they were nonetheless deleterious to mean model performance by between +0.2 to +0.5 % (Table 3). Only [PO$_4$], [NO$_3$], $T$, [O$_2$], $z$, and [Si] consistently improved

the mean ML model, which corresponds to model #3112 (testing MAE of 4.3 nmol kg$^{-1}$).
However, visual inspection of model #3112 output reveals that it does not reproduce expected
near-shore surface plumes of elevated [Ba] close to certain major rivers (see Supplement). Though
volumetrically minor, riverine inputs are a geochemically important component of the marine Ba
cycle, and the existence of nearshore Ba plumes underpins a major proxy application of Ba. Near-
shore riverine influence is easily discerned by low $S$; we thus explored output from model #3080,
which is identical to model #3112, but includes $S$ as a seventh feature during training. Models
#3080 and #3112 exhibit identical statistical performance for the testing data (MAE = 4.3 nmol
kg$^{-1}$; Fig. S1) and make similar predictions for mean marine [Ba] and $\Omega_{barite}$ (89 nmol kg$^{-1}$ and
0.82, respectively; see Supplement). The similar performance of the two models is consistent with
$S$ exerting a near-negligible impact on overall model performance (Table 3). Despite this small
effect, model #3080 is better able to reproduce riverine [Ba] plumes compared to model #3112
(see Supplement). We therefore consider model #3080 to be our best estimate of marine [Ba].
Model #3080 achieves a MAPE of 6.0 %, which represents a 39 % improvement over existing
methods to estimate [Ba] (Fig. 4). We henceforth consider model #3080 as our optimal predictor
model, which we use to simulate [Ba], Ba*, and $\Omega_{barite}$ in Figures 5–9.
**5.2. Model validation**
We now explore the validity of model #3080 in terms of its oceanographic consistency, the sources
of uncertainty that affect its accuracy, and potential limitations of the model output. We find that
model #3080 reproduces the major known features of the marine [Ba] distribution and makes
testable predictions for regions that are yet to be sampled.
*5.2.1. Visual inspection of model output*
Visual inspection of model output is an important component of data analysis considering the
limits of statistical tests (see e.g., Anscombe, 1973). Models may produce statistically satisfactory
fits to the testing data, but the oceanic realism of the output is also important to consider. Modeled
[Ba] should display patterns consistent with related oceanographic properties and exhibit smooth
vertical and spatial variations (Boyle & Edmond, 1975). Predicted [Ba] from model #3080 does
indeed show smooth and systematic spatial and vertical variations that also resembles sparse
observations (Figs. 4–8).
Model #3080 also shows systematic increases in [Ba] close to land, especially near the mouths of
major rivers (Fig. 4). This is reassuring given that elevated sea-surface [Ba] close to rivers is both
widely reported and one of the major proxy applications of Ba: reconstructing spatiotemporal
patterns of terrestrial runoff by measuring the Ba:Ca ratio of carbonates (e.g., Sinclair &
McCulloch, 2004; LaVigne et al., 2016). For example, model #3080 correctly identifies elevated
[Ba] near the Ganges–Brahmaputra (Singh et al., 2013), Río de la Plata (GEOTRACES IDP
Group, 2021), and Yangtze outflows (Cao et al., 2021). Model #3080 also predicts elevated sea-
surface [Ba] in the Gulf of Guinea where several rivers discharge, including the Niger River; the
Eastern Tropical Atlantic associated with the Congo River (Edmond et al., 1978; Zhang et al.,
2023); and in the Gulf of St. Lawrence (St. Lawrence River; see Supplement for additional details
and figures). Except for the Congo River, these predictions of elevated near-shore [Ba] await
corroboration. Interestingly, model #3080 does not predict elevated [Ba] at all major river mouths;
neither the Mississippi nor Amazon Rivers are associated with significant increases in sea-surface
[Ba] (see Supplement). The reasons for the lack of elevated [Ba] near the outflow of these two
rivers is less clear. It is possible that the model is simply inaccurate in these regions, though we
have no particular reason to believe that this is the case. Alternatively, it may reflect seasonal
variations in Ba release that are not captured by our mean annual model (e.g., Joung & Shiller,
2014). It could also indicate that these particular rivers are not major *net* sources of Ba to the
surface ocean, which might be the case if dissolved Ba is being retained in the catchment (e.g.,
Charbonnier et al., 2020) or estuary (e.g., Coffey et al., 1997).
Overall, model #3080 makes accurate, oceanographically consistent predictions of [Ba] in the
Indian Ocean using input data from the WOA. Model #3080 also makes a number of testable
predictions of [Ba] in regions lacking direct observations. Given that these predictions were made
using the same model and the same WOA inputs, we believe that it is reasonable to assume that
model #3080 output is an accurate representation of mean annual global [Ba].

### 5.2.2. Quantifying uncertainties

We now describe and, where possible, quantify two possible sources of uncertainty to our ML model output. Before doing so, we describe how uncertainty is quantified as well as the uncertainty of existing approaches. Certain ML models, such as Gaussian Process Regression, offer low interpretability, meaning it is not possible to assess uncertainty using a conventional error propagation. Thus, all model uncertainties are assessed *post hoc*, by comparing predictions against observations. Our preferred metrics are MAE and MAPE (Eqs. 4, 5). Existing approaches for estimating [Ba] result in a wide range of uncertainties. At the low end, the uncertainty associated with measuring [Ba] in seawater represents a fundamental limit to the accuracy of any model. A number of analysts report measurement uncertainties in the range of 1–2 % (e.g., Pyle et al., 2018; Cao et al., 2020). This level of intra-laboratory uncertainty is typical for [Ba] data obtained using isotope dilution–inductively coupled plasma mass spectrometry, and applies to GEOTRACES-era datasets and to much of the training data from the Indian Ocean. However, intra-laboratory uncertainty is typically much smaller than inter-laboratory uncertainty, which is often between 6–9 % (e.g., Hathorne et al., 2013). At the upper end, the benchmark Ba–Si linear regression achieves a MAPE of 9.7 % in the Indian Ocean (Fig. 4). Thus, useful ML models of [Ba] should achieve MAPE between 1–10 %. Indeed, our favored predictor model, #3080, achieves a MAPE of 6.0 %.

Now we consider two factors that contribute to the observed 6.0 % uncertainty: realization uncertainty and uncertainties in the training data. The realization uncertainty stems from the fact that two models trained on the same training dataset—even with the exact same subset of model features—will produce slightly different predictions. This is due to the holdout cross-folding process used during model training, which partitions the training dataset into random subsets ( Sect. 3.1.). Thus, the training process results in a slightly different trained model each time the model is realized. We quantified the realization uncertainty by training select models 100 times and calculating the relative standard deviation of the different predictions of [Ba] for the 3.3 million values in the output. This uncertainty is small; the median, mean, and maximum realization uncertainty was 0.03 %, 0.04 %, and 0.32 % variability in modeled [Ba].

Next we consider uncertainties in the training data. As noted above, many labs report uncertainties on [Ba] measurements of 1–2 %, while inter-laboratory differences may be up to a factor of five larger. However, this does not consider any uncertainties associated with the other physical and

chemical features used to predict [Ba]. In general, these supporting measurement uncertainties
should be small: all overboard sensors are regularly calibrated and biogeochemical properties in
GEOTRACES are determined using established methods that are based on GO-SHIP best practices
(Hood et al., 2010). Moreover, all GEOTRACES sections include crossover stations that are
intended to facilitate intercalibration of all parameters, including those used here to predict [Ba]
(Fig. 2; Cutter, 2013). The WOA, MLD, Chl. *a*, and bathymetry data products are similarly
subjected to stringent quality review and so we consider it unlikely that these data contribute
systematic biases. We believe that the most likely source of uncertainty relates to the fact that all
predictor information used for model testing in the Indian Ocean was derived from time-averaged
data products, whereas [Ba] was derived from *in situ* measurements. We made this decision
because the *in situ* data were incomplete for all 12 core features (Table 1), and this would have
necessitated interpolation for some features and not others. Since all models were tested using the
same predictor information, the comparison process should avoid systematic errors, though this
does not preclude temporal variability, described next.
*5.2.3. Other considerations*
We now consider four other factors that potentially contribute to the uncertainty of the model
output: short- and long-term temporal variations, limitations of ML, and uncertainties regarding
the thermodynamic properties of $BaSO_4$. Short-timescale variability in [Ba] may affect how
models were evaluated, though this effect is difficult to quantify. In principle, the trained models
should be able to resolve seasonal variations in [Ba] since they were trained on *in situ* physical and
chemical data. In contrast, model predictions in the Indian Ocean were made using annual average
physical and chemical conditions and then evaluated by comparing these predictions against *in*
*situ* [Ba]. The temporal mismatch between Indian Ocean observations and predictions is unlikely
to be significant in the deep ocean, where seasonal variations are minor and the Ba residence time
is longest (e.g., Hayes et al., 2018). Seasonal variations are, however, likely to matter more for the
surface ocean. We were able to minimize some of the impact of these uncertainties by using long-
term averages of Chl. *a* and the maximum monthly mean MLD during model training and testing.
Significant seasonal mismatches for other parameters are unavoidable given that [Ba] data are too
sparse to develop a time-resolved model. We suspect that these variations are most likely to be
significant for boundary sources rather than biogeochemical cycling of Ba; significant
biogeochemical drawdown of surface [Ba] over seasonal timescales appears to be rare (e.g., Esser
& Volpe, 2002), whereas there are large seasonal variations in river discharge that impact near-
shore [Ba] (e.g., Samanta & Dalai, 2016). These suspicions could be tested using a model with
better than $1 \times 1°$ spatial resolution, which—in theory—is possible with model #3080, so long as
similarly high-resolution data are provided for the six predictors utilized by this model ($z$, $T$, $S$,
[O$_2$], [PO$_4$], [NO$_3$], and [Si]). While it is challenging to precisely quantify seasonal uncertainties,
we note that model #3080 performs well at low [Ba], which is found mostly near the surface, where
seasonal variations should exhibit the largest effects. Likewise, seasonal variations will have only
a minor effect on our calculations of global mean [Ba] or $\Omega_{barite}$ (Fig. 8).
Long-term variability in [Ba] may also influence model performance, since the testing data from
the Indian Ocean were collected between 1977 (GEOSECS) and 2008 (SS259; Fig. 2). If secular
changes in Indian Ocean [Ba] were occurring, we might expect models to make accurate
predictions for some datasets at the expense of others. In contrast, we note that model #3080
reproduces all testing datasets similarly well, with the exception of a subset of samples from SS259
in the deep Bay of Bengal. Here we observe that model #3080 predicts 18 % higher [Ba] than
observed by Singh et al. (2013) for the 42 samples between 1,000–3,000 m (Figs. 4B; 7A, B).
Interestingly, model #3080 correctly predicts [Ba] at nearby GEOSECS stations 445 and 446, also
in the Bay of Bengal, sampled some 31 years prior to SS259. We briefly consider three possibilities
for the origin of this regional model–data discrepancy. It may derive from the fact that model
#3080 does not include the features needed to correctly predict [Ba] in these samples. We view
this as the least likely possibility as model #3080 performs well for other samples from the northern
Indian Ocean, including samples shallower than 1,000 m from Singh et al. (2013). Another
possibility is that it could reflect an 18 % decrease in [Ba] in the deep Bay of Bengal since the
GEOSECS survey in the 1970's. Lastly, it could reflect differences in how *in situ* [Ba] was
measured, noting that Singh et al. (2013) opted for standard addition instead of isotope dilution.
We currently lack the data needed to confidently distinguish between these latter two possibilities.
A third factor concerns the limitations of ML itself. We note that no trained model was able to
achieve a MAPE better than ~6 %. This 6 % value may represent one of three things. First, it may
point toward an intrinsic limitation of Gaussian Process Regression. Other types of ML, such as
Decision Trees or Artificial Neural Networks, may be able to achieve superior accuracy, though
this was not investigated. Second, it may indicate that the 12 features investigated provide
insufficient information about [Ba] to achieve higher accuracy. We view this as unlikely given that
our earlier analysis showed that only six–nine features were needed to accurately simulate [Ba]
and that the 12 features tested have proved useful in other studies simulating dissolved tracer
distributions (e.g., Rafter et al., 2019; Sherwen et al., 2019; Roshan & DeVries, 2021). However,
this does not rule out the existence of other features beyond the 12 that we tested that are more
useful for predicting [Ba], only that we did not investigate them. Third, it is possible that the lowest
MAPE of ~6 % reflects the current limit of inter-laboratory uncertainty in determining [Ba]. We
note that inter-laboratory uncertainties of 6–9 % were reported for the measurement of Ba:Ca in
carbonates ($n$ = 10 labs; Hathorne et al., 2013). If the ~6 % MAPE derives from inter-laboratory
uncertainty, it is unlikely that further model refinements will improve the accuracy of [Ba]
predictions: the fundamental limitation is the data, not the model.
A final source of uncertainty concerns the computation of $\Omega_{\text{barite}}$, which contains two further
sources of uncertainty: the thermodynamic model and the solubility coefficients used to calculate
$K_{\text{sp}}$. We calculated $\Omega_{\text{barite}}$ based on the computation described by Rushdi et al. (2000), and our
approach yields similar values to their study and several others (e.g., Jeandel et al., 1996; Monnin
et al., 1999; see Appendix). The model used by Rushdi et al. (2000) is based on $BaSO_4$ solubility
data from Raju & Atkinson (1988), who note good agreement with the thermodynamic data of
Blount (1977). These solubility data were obtained based on experimentation with lab-made,
coarse-grained $BaSO_4$, which is unlikely to be wholly representative of the microcrystalline $BaSO_4$
precipitates found in seawater. Thus, the absolute values of $\Omega_{\text{barite}}$ calculated here may be subject
to eventual revision; however, the vertical (Fig. 1), spatial (Figs. 4–8), and whole-ocean (Fig. 9)
trends in $\Omega_{\text{barite}}$ are robust. Should new thermodynamic data for marine-relevant micron-sized
pelagic $BaSO_4$ become available, updated maps of $\Omega_{\text{barite}}$ could be recalculated using model #3080-
derived [Ba] data. Given the nature of these uncertainties, we opted to calculate prediction
uncertainties for $\Omega_{\text{barite}}$ empirically by comparison to literature data (see Appendix). This yields a
value between 0.08 and 0.10, similar to the 10 % prediction error reported by Monnin et al. (1999).
We can calculate $\Omega_{barite}$ to a high degree of precision; however, there are numerous uncertainties
pertaining to ML-predicted [Ba], the $BaSO_4$ solubility coefficients used to calculate $K_{sp}$, and the
thermodynamic model used in the computation of $\Omega_{barite}$ (Sect. 5.2.). Thus,

## 623   5.3. Barium in seawater: A global perspective

Here we provide an overview of the main model features in [Ba], Ba* and $\Omega_{barite}$, then outline three
possible applications of the model output.

### 626   *5.3.1. Dissolved distribution of [Ba]*

Model #3080 predictions show several interesting features in [Ba] (Figs. 5–8). The model
reproduces the expected nutrient-like distribution of [Ba] (Fig. 1C) and shows a general increase
in [Ba] along the Meridional Overturning Circulation: volume-weighted mean [Ba] increases from
67 to 90 to 106 nmol $kg^{-1}$ from the Atlantic to Indian to the Pacific Ocean, respectively. The model
also predicts some variation in shallow [Ba] that follows major surface-water currents, such as a
region of elevated [Ba] associated with the North Pacific Current, as well as low [Ba] in the western
North Atlantic associated with the Gulf Stream (Fig. 5B; Talley et al., 2011). However, these
features and the processes driving them await corroboration.
Considering the ocean as a whole, we can use our model to calculate the total Ba inventory of
seawater. Using the mean oceanic [Ba] of 89 nmol $kg^{-1}$ and multiplying by the mass of seawater
($1.37\times10^{21}$ kg) yields a total inventory of 122±7 Tmol Ba, whereby the uncertainty is based on the
MAPE of model #3080 (6.0 %). This estimate of the total oceanic Ba inventory is between 11–21
% lower than existing estimates of 145 Tmol Ba (Dickens et al., 2003; Carter et al., 2020). Given
the range of probable global marine Ba fluxes between 18 (Paytan & Kastner, 1996) and 44 Gmol
Ba $yr^{-1}$ (Rahman et al., 2022), our inventory estimate places the mean residence time of Ba in
seawater between 2,600–7,200 years.

*5.3.2. The Ba–Si relationship*

We now quantify spatial and vertical variations in the Ba–Si relationship, which we explore using Ba*. Star tracers, such as Ba*, highlight the processes affecting the distribution of a tracer by comparing it to another tracer that shares the same circulation (Gruber & Sarmiento, 1997). The concept has since been extended to study the processes affecting the distributions of many other bioactive elements, including Si (Si*, relative to N; Sarimento et al., 2004), cadmium (Cd*, relative to P; Baars et al., 2014), zinc (Zn*, relative to Si; Wyatt et al., 2014). First defined by Horner et al. (2015) for Ba, Ba* is analogous to other star tracers: it is a measure of Ba–Si decoupling whereby larger values indicate larger Ba–Si deviations relative to expected mean ocean behavior. Vertical or spatial differences in Ba and Si sources or sinks will drive variations in Ba*, as will any Ba:Si fractionation occuring during their combined cycling. Conversely, if all Ba and Si cycling occurs in the same places (and with a fixed Ba:Si ratio), no Ba–Si decoupling will occur and Ba* will exhibit conservative behavior. Since Ba and Si are cycled by different processes *and* there are large vertical and spatial variations in the intensity of these processes (e.g., Bishop, 1989), significant variations in Ba* are possible. We now explore these variations.

In the surface ocean, patterns of Ba* generally resemble those of [Ba] (Fig. 4). In large parts of the ocean, surface [Si] approaches 0 $\mu$mol kg$^{-1}$; thus, variations in Ba* derive mostly from variations in [Ba]. This is most evident when examining regions with significant terrestrial input of Ba, such as from major rivers (Sect. 5.2.1) and from rivers and continental shelves in the Arctic (e.g., Guay & Falkner, 1998; Whitmore et al., 2022; Fig. 5A). The Southern Ocean also exhibits positive Ba*, though we suspect the mechanism is different. Here we observe a belt of waters with positive Ba* $\approx$+20 nmol kg$^{-1}$ centered on the Polar Frontal Zone—the region between the Antarctic Polar Front and the Subantarctic Front (Orsi et al., 1995; Fig. 5A). Silicic acid is intensely stripped from waters that transit northward through this region (e.g., Sarmiento et al., 2004), potentially contributing to elevated Ba* at the sea surface. Dissolved [Ba] and Ba* then decrease to the north of the Subantarctic front, partly driven by extensive particulate Ba formation in the frontal region (e.g., Bishop, 1989).

At 1,000 m, the Atlantic, South Pacific, and southern Indian Oceans exhibit positive Ba* around +10 nmol kg$^{-1}$, whereas the North Pacific, Southern, and northern Indian Oceans are negative between −10 to −20 nmol kg$^{-1}$ (Fig. 6C). The positive anomalies are likely related to the northward

spreading of southern-sourced intermediate waters that originate within the Polar Frontal Zone and
carry positive Ba* into the low latitudes (e.g., Bates et al., 2017). In the Atlantic, these values are
carried all the way to the north of the basin and return as North Atlantic Deep Water with only
minor modifications to Ba* ($\approx$+10 nmol kg$^{-1}$; Figs. 6C, 7C, 8C). Negative Ba* in the North Pacific,
Southern, and northern Indian Ocean at 1,000 m likely reflects a mixture of hydrographic processes
and *in situ* processes. For example, the extensive region of negative Ba* in the North Pacific is
closely associated with North Pacific Intermediate Water, which originates in the Sea of Okhotsk
(Talley, 1991). While the specific mechanism sustaining this particular Ba* feature is unknown, it
most possibly reflects a combination of preferential removal of Ba relative to Si in the source water
formation region (such as from particulate Ba formation) and weak vertical mixing in the
subsurface North Pacific relative to lateral transports (e.g., Kawabe & Fujio, 2010). We suspect
that the negative Ba* values seen above 1,000 m in the northern Indian Ocean originate through
processes occurring internally within this basin, as the majority of the Indian Ocean below 1,000
m exhibits positive Ba*. A possible mechanism for these shallow negative Ba* anomalies may
relate to the relatively weak overturning transports (Talley, 2008) and strong particulate Ba cycle
north of 30 °S (Singh et al., 2013), though this awaits more detailed investigation.
Lastly, the Southern Ocean exhibits negative Ba* between –10 and –20 nmol kg$^{-1}$ from $\approx$200 m
water depth to the seafloor. These negative anomalies in Ba* appear to be associated with
Circumpolar Deep Water and, below that, Antarctic Bottom Water; the influence of the latter can
also be seen in near-bottom negative Ba* in the South Pacific, southern Indian, and South Atlantic
Oceans (Fig. 8C). As with the other basins, the origin of the negative Ba* waters in the Southern
Ocean likely reflects a combination of *in situ* and circulation-related phenomena. For example, in
the Southern Ocean, Si is only stripped at the very surface, whereas particulate Ba formation is
thought to be greatest in the mesopelagic (i.e., between 200–1,000 m; e.g., Stroobants et al. 1991).
Barite formation is generally considered to be related to the regeneration of particulate organic
matter (e.g., Chow & Goldberg, 1960), whereby the former consumes Ba and the latter releases
Si. Thus, intense organic matter remineralization and associated pelagic $BaSO_4$ precipitation could
contribute to negative Ba* in the mesopelagic Southern Ocean. Similarly, the Si cycle in the
Southern Ocean tends to 'trap' a significant fraction of the global Si inventory in the waters
circulating close to Antartica (e.g., Holzer et al., 2014). Since the calculation of Ba* depends on
both [Ba] and [Si], waters with elevated [Si] will exhibit lower Ba* whether or not there is elevated
Ba removal.
By 2,000 m, almost all of the ocean north of 50 °S exhibits positive Ba* (Fig. 7C). By 4,000 m,
the areal extent of the positive-Ba* waters shrinks to encompass the area north of 30 °S (Fig. 8C).
Despite covering a smaller area, the abyssal ocean exhibits the most positive Ba* values outside
of the surface of the Southern Ocean. The reasons for elevated and increasing Ba* between the
deep and abyssal oceans likely reflects a mixture of local and regional processes, and we offer two
speculative explanations for these patterns. First, Si trapping in the Southern Ocean potentially
renders most of the deep ocean away from Antarctica deficient in Si relative to Ba. Thus, much of
the ocean may exhibit more positive Ba* than the deep circum-Antarctic region due to processes
unrelated to Ba cycling. Second, the most positive Ba* values are generally found close to the
seafloor, rather than the mid-depths, especially in the North Pacific, the Peru and Chile Basins,
and the Philippine Sea. This may indicate a mechanism that preferentially removes Ba (relative to
Si) from the mid-depths, or input of Ba (relative to Si) close to the seafloor.
Systematic variations in Ba* arise due to differences in the marine biogeochemical cycles of Ba
and Si. While, in some cases, the specific drivers of these variations remains unresolved, our model
identifies multiple hotspots of Ba–Si decoupling that warrant additional study.
*5.3.3. Barite saturation state of seawater*
Here we show that our approach can predict $\Omega_{barite}$ with an MAE of 0.08, that our output is in
agreement with published values, and that the deep ocean, below 1,000 m, is at saturation with
respect to $BaSO_4$. By comparison to literature data, we estimate that our model achieves a typical
prediction uncertainty on $\Omega_{barite}$ of 0.08 (see Appendix). Accordingly, values of $\Omega_{barite}$ between
0.92–1.08 can be considered as 'BaSO$_4$ saturated,' whereas values of $\Omega_{barite}$ <0.92 or >1.08 indicate
under- or super-saturation, respectively. Global patterns in $\Omega_{barite}$ derived using our model are
similar to those reported by Monnin et al. (1999) and Rushdi et al. (2000). Readers looking for
detailed basin-by-basin descriptions of $\Omega_{barite}$ are directed to those studies. Briefly our model shows
that, excepting the high latitudes, the surface ocean is undersaturated with respect to $BaSO_4$ (i.e.,
$\Omega_{barite}$ <0.92). The lowest values of $\Omega_{barite}$ in the open ocean are observed in the hot, salty cores of
the Subtropical Gyres ($\Omega_{barite}$ between 0.1–0.2; Fig. 5D). Conversely, the cold and fresh polar
regions exhibit supersaturation at the sea surface, though there are important differences between
the Southern and Arctic Oceans. The Southern Ocean exhibits $BaSO_4$ saturation to depths around
2,000 m, whereas the Arctic Ocean switches to undersaturated conditions below the halocline
(~250 m). At 1,000 m, most of the North Pacific achieves saturation (or slight supersaturation)
with respect to $BaSO_4$ (Fig. 6D) and at 2,000 m almost all of the ocean exhibits $\Omega_{barite}$ >0.92. The
main exceptions to this are the Atlantic Ocean, which is undersaturated at all depths, and the
southern Indian Ocean between 35–50 °S (Fig. 7D). The South Pacific and Indian Oceans return
to undersaturated conditions by 4,000 m, whereas parts of the North Pacific remain saturated to
the seafloor (Fig. 8D). From a global perspective, the oceans are slightly undersaturated with
respect to $BaSO_4$: volume-weighted mean $\Omega_{barite}$ = 0.82; however, the ocean between 1,000–5,500
m exhibits $\Omega_{barite}$ ≥0.92 (Fig. 9). This result implies that the deep ocean, as a whole, is close to
chemical equilibrium with respect to $BaSO_4$.
*5.3.4. Model applications*
In the spirit of maximizing model utility, we suggest three possible uses for model #3080 outputs.
First, the outputs can be used for model intercomparison and intercalibration. For example, a
number of statistical models, such as Optimum Multiparameter Optimization, have been
successfully used to study Ba cycling in the North Atlantic (Le Roy et al., 2018; Rahman et al.,
2022), Southeast Pacific (Rahman et al., 2022), and Mediterranean Sea (Jullion et al., 2017). These
models can apportion the relative contributions of *in situ* biogeochemical cycling and conservative
mixing to observed [Ba]; however, accurate quantification of these processes requires *a priori*
knowledge of end-member water mass [Ba], which model #3080 can provide. Our model could
also be used to benchmark output from process-based models, such as Ocean Circulation Inverse
Models (e.g., John et al., 2020; Roshan & DeVries, 2021). Second, the output can be used for
interpolation purposes. Many groups investigated Ba partitioning into various types of marine
carbonates (see Sect. 1 for examples); however, these investigations are sometimes performed
without a co-located measurement of [Ba]. In these cases output from model #3080 could be used
to help calibrate specific substrates, such as deep-sea corals or benthic forams. This also avoids
the potential for circular reasoning whereby [Si] is used to estimate [Ba], which is then
reconstructed from the Ba:Ca ratio of carbonates to estimate [Si]. Third, the model output makes
testable predictions for regions of the ocean that have yet to be sampled by GEOTRACES-style
surveys. Several of these regions, such as the Southern Ocean, exhibit with sharp lateral and
vertical gradients in [Ba], Ba*, and $\Omega_{barite}$. Such gradients should be considered prime targets for
future process-oriented studies of marine Ba cycling.

## 6. Data availability

Data described in this manuscript can be accessed at the *Biological and Chemical Oceanography*
*Data Management Office* under data doi:10.26008/1912/bco-dmo.885506.1 (Horner & Mete,
768 2023).

## 7. Conclusions

This study presents a spatially and vertically resolved global model of [Ba] determined using
Gaussian Process Regression machine learning. The model reproduces several known features of
the marine [Ba] distribution and makes testable predictions in regions that are yet to be sampled.
Analysis of the model output reveals the mean oceanic [Ba] is 89 nmol kg$^{-1}$, implying a total
marine Ba inventory of 122±7 Tmol. Using predictors from the World Ocean Atlas, we also
estimate the global distribution of Ba* and $\Omega_{barite}$. Both properties exhibit significant gradients that
can be systematically investigated in future studies. The mean oceanic $\Omega_{barite}$ is 0.82, though
between 1,000–5,500 m the mean is ≥0.92, implying that the deep ocean is at equilibrium with
respect to BaSO$_4$. Our model output should prove valuable in studies of Ba biogeochemistry,
specifically for statistical- and process-based model validation, calibrating sedimentary archives,
and for identifying promising regions for further study. More broadly, our study demonstrates the
utility of using machine learning to accurately simulate the distributions of trace elements in
seawater. With minor adjustments, our approach could be employed to make predictions for other
dissolved tracers in the sea.

## Appendix

Here we compare our results with published profiles of $\Omega_{barite}$. Our results were calculated using the thermodynamic model of Rusdi et al. (2000), model #3080 [Ba], and WOA $T$, $S$, and pressure. Literature profiles of $\Omega_{barite}$ were calculated using one of three different thermodynamic models and *in situ* observations of [Ba], $T$, $S$, and pressure. In general, there is strong agreement between modeled and *in situ* $\Omega_{barite}$ whereby our model reproduces the shape of published profiles (Fig. A1). There are, however, some small systematic offsets between the various approaches, and we suspect that these derive from differences in the underlying thermodynamic models.

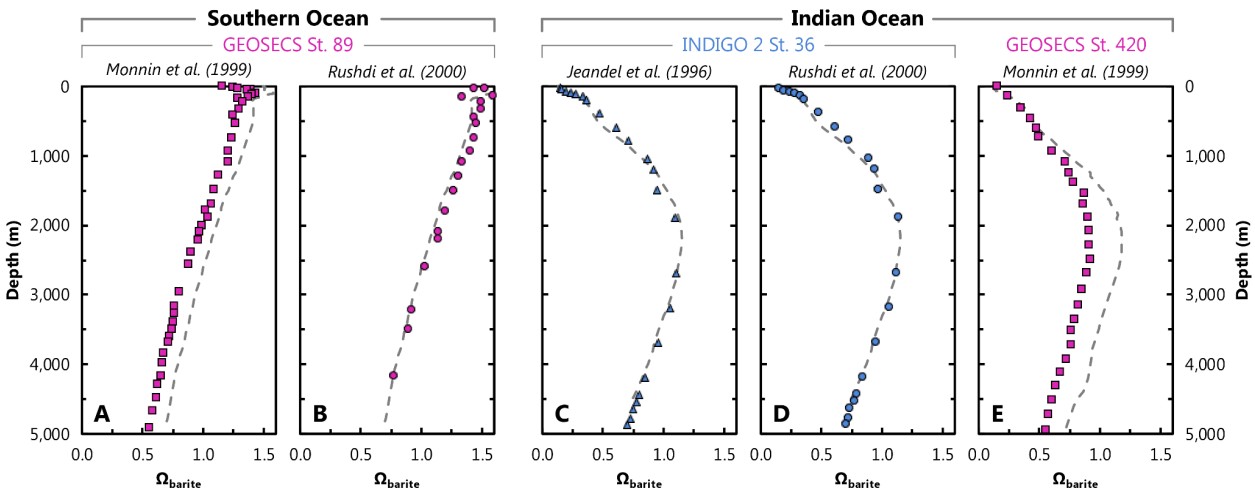

**Figure A1. Comparison of literature- (symbols) and Model #3080-derived (dashed line) values of $\Omega_{barite}$.** Panels **A** and **B** show profiles of $\Omega_{barite}$ at GEOSECS St. 89 (60°0' S, 0°2' E). The other panels are from the Indian Ocean: **C** and **D** are from INDIGO 2 St. 36 (6°9' S, 50°55' E) and **E** from GEOSECS St. 420 (0°3' S, 50°55' E), some ≈675 km north of INDIGO 2 St. 36.

We compare our model output with literature data $\Omega_{barite}$ at two locations in two basins (Fig. A1). These locations were chosen to ensure a fair comparison between studies; at each location, at least two studies calculated profiles of $\Omega_{barite}$ using the same underlying *in situ* data for [Ba], $T$, $S$, and pressure. Thus, any differences in modeled $\Omega_{barite}$ should derive from the thermodynamic model and not the input data. Likewise, literature profiles at these locations were based on calculations for pure, rather than strontian, $BaSO_4$, as in our study. Published profiles of $\Omega_{barite}$ were extracted graphically from each study using *WebPlotDigitizer* (Rohatgi, 2022). This extraction process may

introduce some minor scatter in the literature data, though this is relatively minor compared to the
range of variation in $\Omega_{barite}$.
First, we examine profiles of $\Omega_{barite}$ reported for GEOSECS St. 89 in the Southern Ocean (Fig. A1;
Monnin et al., 1999; Rushdi et al., 2000). Modeled and published profiles show supersaturation in
the surface ocean and undersaturation below 2,000–2,500 m. Profiles from Rushdi et al. (2000)
show excellent agreement with $\Omega_{barite}$ calculated from model #3080 [Ba] and WOA *T*, *S*, and
pressure, with our output offset by a MAE of 0.06 (*n* = 22). Given that we use the same
thermodynamic model as Rushdi et al. (2000), the overall excellent agreement with their study is
not surprising. However, the result is nonetheless reassuring since our study uses mean annual
values for the various inputs, whereas Rushdi et al. (2000) utilized *in situ* data. There is a slightly
larger offset between our profile of $\Omega_{barite}$ and that calculated by Monnin et al. (1999), with our
respective profile exhibiting an MAE of 0.13 (*n* = 41). This most likely reflects differences in the
underlying thermodynamic model and not the *in situ* data since our model reproduces the same
overall profile shape as Monnin et al. (1999). Likewise, both Monnin et al. (1999) and Rushdi et
al. (2000) used the same *in situ* input data and their results are highly comparable, albeit with an
offset similar to that between our results and Monnin et al. (1999).
Next we examine profiles of $\Omega_{barite}$ in the Indian Ocean for samples from INDIGO 2 St. 36 (Fig.
A1; Jeandel et al., 1996; Rushdi et al., 2000). Profiles of $\Omega_{barite}$ show undersaturation at the surface,
moderate supersaturation between 2,000–3,500 m, then return to undersaturated conditions down
to the seafloor. Our profile shows overall excellent agreement with that of Jeandel et al. (1996),
whereby a comparison of $\Omega_{barite}$ yields a MAE of of 0.03 (*n* = 21). Our profile shows similarly
good agreement with Rushdi et al. (2000), whereby a comparison between our respective values
of $\Omega_{barite}$ yields a MAE of 0.04 (*n* = 20).
We also compared our results with data from St. 420 of GEOSECS (Monnin et al., 1999), which
is located ≈675 km north of INDIGO 2 St. 36 (Fig. 2). As with data from the Southern Ocean
(GEOSECS St. 89), our profile data are offset to higher $\Omega_{barite}$ than those of Monnin et al. (1999),
with slightly larger MAE of 0.16 (*n* = 29). However, our modeled $\Omega_{barite}$ is generally in much closer
agreement with Monnin et al. (1999) above 1,100 m than below, equivalent to a MAE of 0.04 (*n*
= 8) and 0.21 (*n* = 21), respectively. In this case it is more challenging to ascribe a unique cause
of the differences in calculated $\Omega_{barite}$; these offsets could relate to differences in the predictors or
the thermodynamic model.
We can use these comparisons to estimate the prediction uncertainty on our model-derived values
of $\Omega_{barite}$. The MAE of the 133 comparisons shown in Fig. A1 yields a value of 0.10. However,
there are different numbers of points in each profile; we thus believe it is more appropriate to
average the MAE calculated for each of the five profiles, which yields a value of 0.08. Both values
are similar to the 10 % prediction uncertainty reported by Monnin et al. (1999).
Overall, our ML-derived profiles of $\Omega_{barite}$ show excellent agreement with *in situ* data, both in
terms of profile shape and values of $\Omega_{barite}$. We use this comparison to estimate the prediction
uncertainty on ML-derived values of $\Omega_{barite}$, which we calculate as being between 0.08 and 0.10.
Should a revised thermodynamic model and/or improved $BaSO_4$ solubility coefficients become
available, a new grid of $\Omega_{barite}$ could be calculated using Model #3080 [Ba] and WOA *T*, *S*, and
pressure data.

## Author contributions

Project conceptualization and funding acquisition by T.J.H. Data curation, formal analysis, investigation, and methodology by O.Z.M., A.V.S., H.H.K, and T.J.H. Data visualization by A.V.S. and T.J.H. Software provided by O.Z.M., A.V.S., H.H.K., and A.G.D. Writing (original draft) by O.Z.M. and T.J.H.; review and editing by A.V.S., H.H.K., A.G.D., L.M.W., A.M.S., M.G., and W.D.L.

## Competing interests

The authors declare that they have no conflict of interest.

## Acknowledgements

We are thankful to the many data originators who contributed dissolved Ba data to the 2021 GEOTRACES Intermediate Data Product, as well as the funding agencies that made those contributions possible. The GEOTRACES IDP represents an international collaboration and is endorsed by the Scientific Committee on Oceanic Research. We are especially grateful to Frank Dehairs, who provided comments on an early draft of the text and shared additional testing data from the Indian Ocean, as well as Karen Grissom, who provided laboratory assistance to A.M.S. We kindly acknowledge use of the *Discovery* high-performance compute nodes at Dartmouth College Research Computing. We are grateful to the Editor, Christophe Monnin, Frank Pavia, and two anonymous reviewers who provided insightful and constructive comments that helped us improve the study.

## Financial support

O.Z.M was supported by WHOI's Academic Programs Office through a *Summer Student Fellowship*. A.M.S. acknowledges support from the U.S. National Science Foundation (OCE-0927951, OCE-1137851, OCE-1261214, OCE-1436312, and OCE-1737024), as does T.J.H. (OCE-1736949 and OCE-2023456). T.J.H. was further supported by *The Andrew W. Mellon Foundation Endowed Fund for Innovative Research* and *The Breene M. Kerr Early Career Scientist Endowment Fund*.

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
