# Peer review of "Barium in seawater 1"

_Earth System Science Data, 2023_

## Author Comment (AC1)

**Response to Reviewer Comments**

**Barium in seawater: Dissolved distribution, relationship to silicon, and barite saturation state determined using machine learning**

We thank the four reviewers for their insightful and helpful comments on our study. We think these comments have greatly improved the revised manuscript. Below we reprint all reviewer comments (in black). Our responses follow (in blue) and any corresponding changes to the manuscript are shown in green.

**RC1: 'Comment on essd-2023-67'**

**Anonymous Referee #1**

*Major comments*

The manuscript "Barium in seawater: Dissolved distribution, relationship to silicon, and barite saturation state determined using machine learning" by Mete et al. developed a Gaussian Progress Regression Machine Learning (ML) approach to predict dissolved Ba ([Ba]) in the ocean. This study is significant for understanding the marine Ba cycle because it provides a global picture of the vertical and spatial distribution of [Ba] and $\Omega_{barite}$ and suggests factors that intimately link to [Ba]. It is exciting that the ML-derived Ba profiles are in excellent agreement with *in situ* data. The manuscript is well-reasoned and well-written. I enjoy reading it and am happy to recommend it for publication if the following concerns could be addressed. I hope the authors find my comments constructive and help them make the manuscript more impactful.

We are delighted to read that the reviewer finds the study well put together. We found these comments especially helpful and made a number of changes in response to them.

Sect. 2 and 3.1: The ML model split the observed datasets into two partitions: the data from the Arctic, Atlantic, Pacific, and Southern Oceans were used for model training, whereas the data from the Indian Ocean were reserved for model testing. Yes, as indicated by the authors, the location-based training-testing separation is to minimize overfitting. However, we also need to be careful that the training data happen to perfectly cover the minimum and maximum [Ba] (according to Figs. 4A-7A), so [Ba] in the Indian Ocean is very well predicted. I would like to know whether the ML model also works well when testing data fall outside training data. That said, it is necessary to include the randomly assigned training-testing separation results for comparison in the appendix.

These are great points and we've made three changes to the study to address them. Please note that in response to a comment raised by Reviewer #4, we now identify model #3080 as the optimal predictor model. This model is referenced in place of model #3336 in subsequent discussion and figures.

First, we note in the main text that the ML model output is within the ranges encountered during training:

"The range of model #3080 predictions is within the range of [Ba] encountered in the 4,345 training data (17.1–159.8 nmol kg–1). This is an important consideration when assessing the accuracy of Gaussian Process Regression models, and we provide additional discussion of this point in the Supplement."

Second, we investigated how restricting the range of the training data would affect model performance. The following text and figures have been added to the supplement:

"Gaussian Process Regression (GPR) models are highly adept at making accurate geospatial predictions of a target variable, particularly when the training data contain a certain level of noise. However, GPR models are oftentimes less accurate than other methods when making predictions beyond the ranges encountered during training (Cressie, 1993). To investigate whether our preferred predictor model was subject to similar biases, we analyzed the performance of model #3080 when provided with a narrower range of training data. This meant restricting the range of [Ba] values seen during model training and comparing these outputs against those generated by model #3080 when trained on the full training dataset.

"To achieve this, we identified the bottom (51.4) and top (98.9 nmol kg–1) sextile in the Indian Ocean testing data and removed all [Ba] observations from the training data that were outside of this range (i.e., only samples with [Ba] between 51.4–98.9 nmol kg–1 were included). This reduced the number of [Ba] observations in the training data from 4,345 to 2,295. We then retrained model #3080 (z, T, S, [O2], [PO4], [NO3], and [Si]) on these 2,295 data and used this retrained model (hereafter model #3080N) to predict [Ba] for the Indian Ocean and on a global basis. A comparison of [Ba] predictions made using model #3080N and from model #3080 are shown below in Figs. S9 and S10."

[Figure]

**Figure S9. Comparison of model #3080N and model #3080 outputs for the Indian Ocean testing data**. Model #3080 was trained using a narrowed version of the training data compared to model #3080, which saw the full training database.

[Figure]

**Figure S10. Boxplot of [Ba] values for training, testing, and ML model predictions**. Each box shows a five-number summary for the relevant dataset: median (horizontal line), the 75th and 25th percentiles (top and bottom of box, respectively), and the maximum and minimum non-outlier values (upper and lower whiskers, respectively). Statistical outliers are indicated by '+'. Boxes labeled 'observations' summarize the *in situ* data used in model training and testing, respectively. The next two boxes show model #3080 predictions for the global ocean and the testing data. The final three boxes show the distribution of [Ba] values in the 'narrow' training data and the resultant spread of [Ba] predicted by model #3080N, which was trained using only these data.

"This analysis shows that model #3080N reproduces the median and interquartile range of the Indian Ocean testing data (Figs. S9) as well as for the global predictions (Fig. S10). Likewise, model #3080N can predict values of [Ba] outside of the ranges encountered in model training, but only by between 5–10 %. As such, model #3080N underestimates the true range of [Ba] values seen in the ocean and achieves a lower overall accuracy of [Ba] predictions compared to model #3080 (#3080N MAPE = 8.8 % vs 6.0 %; Fig. S9). We conclude that the output from model #3080, and likely other models, is most accurate when it falls within the range of [Ba] encountered during training. Since model #3080 was trained on [Ba] data spanning 17.1–159.8 nmol kg–1, the entire range of model #3080 predictions (26.2–156.8 nmol kg–1; Fig. S10) falls within the range seen during training. Thus, we conclude that the results from model #3080 are generally robust as the model did not extrapolate beyond the range of [Ba] encountered during training."

Third, we included results from the random holdout process during model training. This was a really great suggestion and we felt that addressing it required adding a new figure (Fig. 3) and a table (Table 3). Two other reviewers raised related points; rather than duplicating our responses,

we note here that a detailed description of our changes is provided in response to Reviewer #3 and Reviewer #4.

For the paper to benefit the community, additional discussion about the implications of existing interpretations that rely on Ba* would be of great interest. Unlike [Ba] or $\Omega_{\text{barite}}$, the scientific significance of Ba* is not clear in the current version of the manuscript. Specifically, what do the positive and negative Ba* mean? Does the global Ba* heterogeneity in Figs. 4-7 reveal oceanographic and biogeochemical processes affecting the dissolved Ba-Si relationship? I believe the relationship to silicon is one of the main targets of this study which requires in-depth discussion.

This is a great suggestion and we are delighted to have the opportunity to provide some in-depth analysis of Ba*. We made two main changes in response to this comment. First, we described the significance of 'star tracers' in Section 3.:

"A selection of the most accurate models of [Ba] were then used to simulate Ba* and $\Omega_{\text{barite}}$. Star tracers, such as Ba*, are valuable for illustrating processes that influence the cycling of elements in the ocean. First defined for N–P decoupling (N*; Gruber & Sarmiento, 1997) star tracers show variations whenever there are differences in the sources and sinks of the two elements being compared. If there are no differences in sources and sinks, the tracer will show conservative behavior because both elements share the same circulation. Barium-star is based on Ba–Si decoupling and was first defined by Horner et al. (2015). …"

Second, we provided a high-level 'walkthrough' of the Ba* distribution in a new subsection of the Discussion (Sect. 5.3.2.). The text that appears in this new subsection appears below:

[revised manuscript text omitted]

Sect. 5.1: When the authors identified the optimal predictor model, they eliminated features that offer the least improvement to ML model performance. Why are only MLD and chlorophyll a eliminated, but not salinity (they improve the model equivalently low, i.e., -3%)? Including salinity tends not to change the MAD much due to its high p-value. The authors need to justify it further.

This is a really helpful point, and it speaks to an issue raised by Reviewer #4. In response to this comment, we revisited the feature significance analysis (Sect. 4.1.). to explore the effect of different model features on model performance in three steps: (i) for the random holdout cross folding of the training data; (ii) for the regional cross-validation using the testing data; and, (iii) for the regional cross-validation testing data, but only considering the 1,687 models that performed better than the Ba–Si linear regression. We focus our exploration on the third group since only these models perform better than existing methods for predicting [Ba]. The new table (Table 3) and a summary of this analysis appears below:

[revised manuscript text omitted]

We also added and series of visual comparisons of model #3112 and #3080 outputs to the Supplement. An example of one of these comparisons is shown below. The idea here is to illustrate that although model #3112 performs statistically similarly to model #3080, it misses riverine Ba inputs. Since *S* is the only difference between these two models, we conclude that *S* must be important for getting this feature right.

[Figure]

**Figure S1. Comparison of model #3112 and model #3080 outputs for the Indian Ocean testing data**. Model #3112 was trained using six features: *z*, *T*, [O₂], [PO₄], [NO₃], and [Si]. These are the same features as in model #3080, minus *S*. The statistical performance of the two models is highly similar, though model #3112 misses important geochemical features, discussed in the text.

[Figure]

**Figure S2. Seawater chemistry in the Bay of Bengal and Andaman Sea.** Left and center panels show [Ba] at the sea surface from model #3080 and #3112, respectively. Right panel shows sea-surface salinity.

*Minor comments*

L81: The solubility product Ksp is a constant at a given temperature and pH. Thus, Ksp values at different depths are different. The text needs to clarify how [SO4] and Ksp are assigned.

Great point. We now explain in more detail how we determined $\Omega_{barite}$:

"Values of $\Omega$barite were computed using the method described by Rushdi et al. (2000), summarized in Equation 3. The numerator, Q, represents the in situ Ba and sulfate ion product and, in this formulation, depends only on [Ba] and [SO42–] molality. The denominator, Ksp, depends on T, S, and z (i.e., pressure) and is calculated in two steps: in situ T and S are used to calculate the stoichiometric solubility product and then this value is modified by calculating the effect of pressure on partial molal volume and compressibility, which are functions of T and z. As with the calculation of Ba*, values of [Ba]in situ were obtained from ML models and co-located data for T, S, and z were extracted from the WOA (Locarnini et al., 2018; Zweng et al., 2018). Sulfate concentrations were assumed to be conservative with respect to S using [SO42–] = 29.26 mmol kg–1 when salinity = 35 PSU. This latter assumption likely breaks down in certain environments (e.g., where [SO42–] reduction occurs); as such, our model is not used to predict $\Omega$barite in restricted basins, such as the Black Sea or Caspian Sea."

L266-267: [Si]$_{in\ situ}$ and [Ba]$_{in\ situ}$ from the WOA, [Ba]$_{predicted}$ from ML model output?

Sort of; [Si]$_{in\ situ}$ was from the WOA, [Ba]$_{predicted}$ calculated from [Si]$_{in\ situ}$, and [Ba]$_{in\ situ}$ from ML. We've changed the wording here since the original was confusing. The new text reads:

"The global distribution of Ba* was determined by calculating [Ba]predicted (Eq. 2) using [Si]in situ from the WOA 2018 (García et al., 2018b). The values of [Ba]in situ was taken from the ML model output and [Ba]predicted was subtracted from this to yield Ba* (Eq. 1)."

Fig.3B and L502-513: The authors attribute the deviation between observed and ML-modeled [Ba] from SS259 in the deep Bay of Bengal to the uncertainty of *in situ* [Ba] measurements. Could this deviation result from the factors eliminated from Model #3336? This possibility needs to be discussed at least.

A good third option that we hadn't considered (also suggested by Reviewer #3). This paragraph now closes like this:

"We briefly consider three possibilities for the origin of this regional model–data discrepancy. It may derive from the fact that model #3080 does not include the features needed to correctly predict [Ba] in these samples. We view this as the least likely possibility as model #3080 performs well for other samples from the northern Indian Ocean, including samples shallower than 1,000 m from Singh et al. (2013). Another possibility is that it could reflect an 18 % decrease in [Ba] in the deep Bay of Bengal since the GEOSECS survey in the 1970's. Lastly, it could reflect differences in how in situ [Ba] was measured, noting that Singh et al. (2013) opted for standard addition instead of isotope dilution. We currently lack the data needed to confidently distinguish between these latter two possibilities."

**RC2: 'Comment on essd-2023-67'**

**Christophe Monnin**

L. 348 et seq. The criterion for equilibrium cannot be $\Omega$ =1.0000000…. which never happens. Instead a range of $\Omega$ values must be defined, that reflects the uncertainties in the input data (Ba and SO4 concentrations) and in the thermodynamic model (barite solubility product and activity coefficient). Taking $\Omega$ values between 0.9 and 1.1 is already a very demanding criterion that we have retained in our paper (Monnin et al., 199). So the discussion in this paragraph should be corrected according to this. For example $\Omega$ = 0.97 can be considered as a sign of barite equilibrium. See Monnin et al., 1999 for a discussion. I see that the authors discuss this point in section 5.2.3.

We thank Christophe Monnin for his important insights, which were also entertaining to read!

This is an interesting point and we've made five changs in response to this comment. First, we note that the values of $\Omega$ possess an uncertainty (Sect. 4.2):

"Model #3080 contains 3,302,570 predictions for each of [Ba], Ba*, and $\Omega$barite (Sect. 6). Assuming that the MAPE and MAE are good estimates of the prediction error, we estimate that modeled [Ba] and Ba* have uncertainties of 6.0 % and 4.3 nmol kg–1, respectively. Uncertainties on $\Omega$barite were estimated by comparison to literature data, which yields a MAE of 0.08. These estimates are discussed in more detail in Section 5.2 and the Appendix."

Second, also in Section 4.2., we make a similar point when describing the volume-weighted histograms:

"Lastly, $\Omega$barite tends to increase between the 0–250 m, 250–1,000 m, and 1,000–2,000 m depth bins, increasing from 0.42, to 0.65, and 0.96, respectively. Average $\Omega$barite in the deepest bin (2,000–5,500 m) is slightly lower, with a mean value of 0.92 (Fig. 9D). Given the accuracy of our model-derived $\Omega$barite predictions (0.08 to 0.10), the ocean between 1,000–5,500 m is within uncertainty of BaSO4 equilibrium."

Third, we clarify where these uncertainties come from in Section 5.2.3.:

"Given the nature of these uncertainties, we opted to calculate prediction uncertainties for $\Omega$barite empirically by comparison to literature data (see Appendix). This yields a value between 0.08 and 0.10, similar to the 10 % prediction error reported by Monnin et al. (1999)."

Fourth, we provide these details in the Appendix:

"We can use these comparisons to estimate the prediction uncertainty on our model-derived values of $\Omega$barite. The MAE of the 133 comparisons shown in Fig. A1 yields a value of 0.10. However, there are different numbers of points in each profile; we thus believe it is more appropriate to average the MAE calculated for each of the five profiles, which yields a value of 0.08. Both values are similar to the 10 % prediction uncertainty reported by Monnin et al. (1999)."

And fifth, we note what this means for assessing equilibrium in Section 5.3.3.:

"By comparison to literature data, we estimate that our model achieves a typical prediction uncertainty on $\Omega$barite of 0.08 (see Appendix). Accordingly, values of $\Omega$barite between 0.92–1.08 can be considered as 'BaSO4 saturated,' whereas values of $\Omega$barite <0.92 or >1.08 indicate under- or super-saturation, respectively."

The updated Figure A1 (with model #3080 output is shown below):

[Figure]

**Figure A1. Comparison of literature- (symbols) and Model #3080-derived (dashed line) values of $\Omega_{barite}$.** Panels **A** and **B** show profiles of $\Omega_{barite}$ at GEOSECS St. 89 (60°0' S, 0°2' E). The other panels are from the Indian Ocean: **C** and **D** are from INDIGO 2 St. 36 (6°9' S, 50°55' E) and **E** from GEOSECS St. 420 (0°3' S, 50°55' E), some ≈675 km north of INDIGO 2 St. 36.

Also the statement in the abstract that "the ocean below 1,000 m is, on average, at or near saturation" is a simplification of what has been previously depicted by Monnin et al. (and by Rushdi et al.). For example we wrote that in the Pacific Ocean "There is a return to undersaturation of the water column at depths of about 3500 m in the Pacific and of about 2500 m in the Southern Ocean. The reverse is found for GEOSECS station 446 in the Gulf of Bengal for which the highest Ba concentrations can be found at depth: surface waters are undersaturated and equilibrium is reached below 2000 m". This simplifying statement by the authors deteriorates the conclusions that they have obtained with their powerful and elaborate approach.

Fair enough! We've addressed this point in two ways. First, we note in Section 5.3.3. that earlier studies already discerned the major contours in the global distribution of $\Omega$:

"Global patterns in $\Omega$barite derived using our model are similar to those reported by Monnin et al. (1999) and Rushdi et al. (2000). Readers looking for detailed basin-by-basin descriptions of $\Omega$barite are directed to those studies."

Second, we added a caveat in the abstract stating that there are regional variations in $\Omega$:

"We also calculate the saturation state of seawater with respect to barite. In addition to revealing systematic spatial and vertical variations, our results show that the ocean below 1,000 m is at equilibrium with respect to barite."

The discussion averaging barite saturation for the global ocean (L. 579-580:" the ocean below 1,000 m exhibits a mean $\Omega$ barite $\geq 0.92$, which implies that much of the deep ocean is close to saturation with respect to BaSO4") tends to hide the fact that specific regions of the global ocean do not fit in this picture and should be given close attention (e.g. the role of hydrothermal activity above ridges, or, as discussed by the authors, the continental input through river discharge).

We opted to avoid discussing detailed regional patterns in $\Omega$ as the reviewer rightly notes that this has been tackled by previous studies. (Conversely, no such discussion of global variability in Ba* exists, and we took the opportunity provided by Reviewer #1 to address this issue.) In response to this point, we tried to make it clearer that our focus, with respect to $\Omega$, was on three main things: (i) accuracy of our $\Omega$ estimates, (ii) whether our values reproduce known features in $\Omega$, and (iii) the global mean $\Omega$. The following text now appears as the preamble to Section 5.3.3.:

"Here we show that our approach can predict $\Omega$barite with an MAE of 0.08, that our output is in agreement with published values, and that the deep ocean, below 1,000 m, is at saturation with respect to BaSO4."

This being said, the paper is quite well organized, the presentation of the model and of the results quite concise.

Excellent!

No changes made.

Although my opinion is that the discussion repeats in part what has been concluded from what the authors call mechanistic models and as such should be simplified, the paper presents a very good account of the Ba problem in the ocean and a way to address it (by what could be called "brute force"…). It could be published as it is. The fact that the model can be adapted for other tracers with a minimal effort is quite encouraging.

Excellent – thank you! We may even borrow some of this language for use in future presentations: "Barium by brute force: Results from an ML model"

No changes made.

L. 67: missing figure number

Added cross reference to Figure 1.

Typo in the vertical axis of Fig. 4, 5 and 6D: replace the lower 1.75 by 0.75.

Changed.

L.338. Sentence construction inadequate.

Good catch. The revised sentence reads:

"Based on our formulation (Eqs. 1, 2), Ba* varies from $-27.2$ to $+27.9$ nmol kg$-1$ and possesses an unweighted mean of $+2.4$ nmol kg$-1$."

**RC3: 'Comment on essd-2023-67'**

**Frank Pavia**

I really enjoyed reading this paper by Mete et al. The manuscript was well-written, well-organized, extremely clear, and the work generates a product that should be used by chemical oceanographers and paleoceanographers alike. I have a few comments, and only the first is substantial.

We are delighted to read such an encouraging and positive response from Frank Pavia. We found these comments very helpful.

The decision to only use Indian Ocean data as the validation dataset is definitely curious, and I think not justified well-enough in the text. The authors cite Rafter et al. 2019 as their source for doing location-based separation of training and test data to avoid overfitting. Rafter et al., however, don't isolate a single basin for this - they use whole ship transects as their witheld data, and these transects span multiple basins, hemispheres, and latitudes. Testing a globally-trained dataset on a regionally-confined subset of data doesn't, at least to a reader not well-versed in these sorts of choices, inspire the maximum amount of confidence in the results of the global output of the model. Perhaps the authors could more completely explain this choice to bulwark against this criticism.

This is a great point and we've taken the opportunity to clarify our reasoning. Before addressing it, however, we want to note a related point raised by Reviewer #1, who wanted us to include the randomly assigned training–testing separation results for comparison. We now do so in Figure 3 (below), which shows that the model training process vastly underestimates the error in [Ba] predictions. The main reason for this, stated on p. 108 of Rasmussen & Williams (2006) is:

> "[T]he training error is usually a poor proxy for the generalization error, since the model may fit the noise in the training set (over-fit), leading to low training error but poor generalization performance."

This means that if we partition the data randomly we will get small, but unrealistic errors. If we partition geographically we get larger, but more-realistic errors. We now note this in Section 3.2.:

"A significant problem in supervised ML, and particularly Gaussian Process Regression learning, is overfitting: models may fit the noise in the training data, leading to poor generalization performance (Rasmussen & Williams, 2006). Since our goal was to develop a global model of [Ba] using regional training data, we deemed it especially important to identify generalizable models. Generalizable models were identified through a testing process involving regional cross-validation; each trained model was used to predict [Ba] for the 1,157 samples from the Indian Ocean and model predictions were again compared against observations. Importantly, no [Ba] data from the Indian Ocean were seen by any of the models during training. This process helped to identify models that may have been overfit to the training data and can further be used to calculate generalization errors (Sect. 4.1)."

[Figure]

**Figure 3. Effect of feature addition on ML model accuracy.** Accuracy was quantified for each of the 4,095 trained models and quantified here using MAE (note log scale, which differs between panels). The accuracy of trained models is shown for random holdout cross-validation during training (top) and for regional cross-validation during testing (bottom). Square indicates the performance of our favored predictor model, #3080 (see Fig. 4, Sect. 5.1). The accuracy of the Ba–Si linear regression benchmark is shown as a dashed line in the lower panel (MAE = 6.8 nmol kg$^{-1}$). To illustrate data density, points have been randomly positioned within their respective bin and plotted with 80 % transparency.

To further address Reviewer #3's point, we added some discussion to Section 5.1. explaining why we opted for regional cross-validation. We think that these arguments are now much stronger than in the original submission and we appreciate the reviewer's insights.

"Choosing a single, optimal model configuration is challenging given the sheer number of skillful ML models. Below we winnow the list from 4,095 to a single model (#3080). We base our winnowing primarily on the results of the regional cross-validation performed in the Indian Ocean, rather than from the errors determined from random holdout cross folding of the training data. We believe that there are three strong reasons for winnowing in this way. First, Gaussian Process Regression Learners tend to fit the noise in the training data, meaning that the training error is significantly lower than the generalization error (Rasmussen & Williams, 2006). Indeed, trained models showed overall lower performance during testing compared to training, which we believe is evidence of overfitting (Fig. 3, Table 3). Second, a generalizable global model should be able to make predictions in regions where it has not already learned anything about the target variable. Our regional cross-validation approach satisfies this consideration since no Indian Ocean data were seen during model training. Third, the Indian Ocean is an ideal basin for testing as it exhibits the full diversity of features expected to influence [Ba] (riverine inputs, oxygen-minimum zones, coastal upwelling, etc.) and constitutes ≈20 % of the global ocean volume. Likewise, the Indian Ocean captures most of the range in [Ba] seen elsewhere in the ocean (Fig. 9); this likely reflects the input of Atlantic waters through the Aughulas leakage, transport of old Pacific waters via the Indonesian Throughflow, and northward spreading of mode and intermediate waters from the Southern Ocean. We thus assume that the Indian Ocean testing errors are a good approximation of the generalization error, which we now use to winnow the list of models."

Figures 4-7: The 0.75 value for barite saturation is labeled as 1.75 in all these figures.

Good catch (thanks also to Reviewer #2!).

Changed.

Line 342: Change Look to Looking

Done. Now reads:

"Looking at the ocean as a whole, the probability density function of [Ba] roughly resembles a uniform distribution, with a mean ocean [Ba] of 89 nmol kg–1 (Fig. 9A)."

Figure 8: I think it would be helpful to have a key/legend for the basins in A and C, similar to the key for depths in B and D. It is not easy to match the text color of the basins to the colors of the histograms in panels A and C.

Good suggestion. These have been added to the updated histograms for model #3080:

[Figure]

**Figure 9. Stacked, volume-weighted histograms showing the relative frequency distribution of dissolved [Ba] (A, B) and $\Omega_{barite}$ (C, D) in the global ocean.** The left column shows data grouped by basin and the right column shows data grouped by a prescribed depth bin. Numbers in each panel display the mean property value for that bin. Dashed line shows the global mean.

Lines 507-513: Did all or many of the best models tend to produce the same systematic mismatch between predicted and measured Ba for the Singh et al. 2013 data? It would be helpful to know to make sure it isn't a quick [quirk?] of this specific model.

Reviewer #1 had a similar question and we added a few sentences in Section 5.2.3. to clarify this point (reprinted in response to Reviewer #1's comment).

Overall, we don't think it's a specific quirk of this model because several other good models reproduced this offset; indeed, our new 'optimal' predictor model shows similar behavior as model #3336 (the optimal model in the original submission). However, we did not investigate whether every model predicted this mismatch because our focus was on making a globally (rather than regionally) accurate model of [Ba]. It's worth noting that model #3080 does well on the shallow samples from Singh et al. (2013), implying that it's something specific to the deep Bay of Bengal, namely: changes in the deep dissolved Ba inventory since GEOSECS, or inaccuracies with the measurements. Since we cannot distinguish between these two possibilities without more recent samples from the region, we have left it open ended.

Section 5.3. I really enjoyed reading this section and am looking forward to the community's use of this data product.

Excellent!

No changes made.

**RC4: 'Comment on essd-2023-67'**

**Anonymous Referee #4**

*Summary*

This study uses a machine learning approach to reconstruct global Ba concentrations in the ocean, and uses the model output to calculate Ba* and barite saturation state in the global ocean. In general this is solid study that provides model output that will be useful to other researchers, and the methodology is sound, with one exception that I detail below. I think that with minor revisions the study should be acceptable for publication.

We're pleased to read that the reviewer felt that this was a solid study and we are grateful for their comments, which have improved the contribution.

*Specific Comments*

- Line 89: I disagree that mechanistic modeling should be called the "gold standard". A model is useful if one can learn something from it, period. Some mechanistic models are useful, some statistical models are useful.

Fair enough. We took out this language. The new sentence reads:

"In mechanistic or process-based modeling, model outputs are derived from sets of underlying equations that are based on fundamental theory. As such, mechanistic model outputs can be interrogated to obtain understanding of processes and their sensitivities."

- Line 104: The entire process and methodology of this study seems to owe a large intellectual debt to ML-based trace metal modeling studies of Roshan et al. These pioneering studies should be acknowledged here, e.g. Roshan et al. (2018), Roshan et al. (2020)

We are happy to acknowledge these earlier studies. The new sentence reads:

"Machine learning is increasingly being used to solve problems in Earth and environmental sciences, including simulating the dissolved distribution of tracers in the sea (e.g., for cadmium, Roshan & DeVries, 2021; copper, Roshan et al., 2020; iodine, Sherwen et al. 2019; nitrogen isotopes of nitrate, Rafter et al., 2019; and zinc, Roshan et al., 2018)."

- Line 196: Explain what you mean by "non-parameteric" and "kernel-based"

Excellent suggestion. We added a sentence to clarify this:

"This particular ML algorithm is non-parametric, kernel-based, and probabilistic, which means that it does not make strong assumptions about the mapping function, can handle nonlinearities, and takes into account the effect of random occurrences when making predictions."

We also added the following to make clear why we used GPR:

"Gaussian Process Regression algorithms are widely used in geostatistics, where it is often referred to as 'kriging' (e.g., Cressie, 1993; Rasmussen & Williams, 2006; Glover et al., 2011). This type of algorithm is ideal when working with continuous data that also contains a certain level of noise, such as from measurement uncertainty or oceanographic variation."

- Line 196: What is the specific MATLAB function, and what options did you specify

An excellent idea. We now name the function in the main text:

"The MATLAB function, `fitrgp`, was used for model training."

We also note the following:

"A full list of the parameter selections used in `fitrgp` is provided in Table S1."

We then provide a table in the Supplement (Table S1) that explains all the function options, what they do, the value we selected, and why we chose that value. This table is reprinted below:

**Table S1. Function parameters specified for the function used to train ML models.** The MATLAB function `fitrgp` was used to perform model training (MathWorks, 2023). Each option, its purpose, the value assigned, and a justification for the value chosen are shown.

| Option | Description of option | Value selected | Description of the value selected |
|---|---|---|---|
| Fit Method | Method to estimate parameters of the GPR model | `'sd'` | Subset of data points approximation (i.e., selects a smaller subset of training data points and computes the inverse of the covariance matrix only for that subset, while the remaining data points are used to estimate the hyperparameters of the model.) |
| Basis Function | Explicit basis in the GPR model | `'constant'` | H=1 (n-by-1 vector of 1s, where n is the number of observations, i.e., sets the mean of the GPR model to be a constant value, which is equal to the mean of the training output data and is applied to all observations |

| | | | in the training data |
|---|---|---|---|
| Beta | Initial value of the coefficients | | Inferred from the data, thus changes with each run. |
| Sigma | Initial value for the noise standard deviation of the Gaussian process model | `std(y)/sqrt(2)` | Depends on the response data, thus changes with each run. |
| Constant Sigma | Constant value of Sigma for the noise standard deviation of the Gaussian process model | `false` | allows the noise standard deviation to vary across different input points |
| Sigma Lower Bound | Lower bound on the noise standard deviation | `1e-2*std(y)` | Depends on the response data, thus changes with each run. |
| Categorical Predictors | Categorical predictors list | logical vector of length p where each element is false and p is the number of predictors | None of our predictors are categorical. |
| Standardize | Specify whether or not the data should be standardized using mean and standard deviation | `true` | When true, each predictor is centered and scaled to have a mean of zero and a standard deviation of unity. |
| Kernel Function | Form of the covariance function | `'exponential'` | sets an exponential kernel function (i.e., a type of radial basis function that computes the similarity or covariance between two input vectors based on their distance or proximity in the input space) to be used to model the covariance between the input variables. |
| Distance Method | Method for computing inter-point distances | `'fast'` | e.g., $(x-y)^2$ is computed as $x^2 + y^2 - 2*x*y$ when the distance method is fast. |
| Active Set | When specified, the active set indicates the observations to be used in model training. If the active set is predetermined, ActiveSetSize and ActiveSetMethod are not used. | [] | We do not assign a predetermined active set and let the model chose a random active set |

| | | | |
|---|---|---|---|
| Active Set Method | selection method for the Active Set | `'random'` | random selection of active set |
| Random Search Size | Random search set size | `59` | MATLAB default value |
| Tolerance Active Set | Relative tolerance for terminating active set selection | `1e-6` | Controls the convergence tolerance level for the active set algorithm used in the "subset of data points" fitting method. |
| Predict Method | Method used to make predictions | `'exact'` | Specifies that the exact method should be used to make predictions with the trained GPR model |
| Optimizer | Optimizer to use for parameter estimation | `'quasinewton'` | Sets a quasi-Newton method (i.e., a gradient-based optimization algorithm) to estimate the hyperparameters or other parameters of the GPR model. |
| Initial Step Size | Initial step size | `[]` | Empty. Initial step size is not used to determine the initial Hessian approximation. |
| Holdout | A cross-validation method where a fraction of the data is used for validation. | `0.2` | Use 20% of training data for validation and 80% for training. |

- Line 199: Explain the meaning of "basis" and "kernel-function" parameters

These are now all described in Table S1 (above). We think that this change makes the main text simpler to follow and our methods easier to replicate.

- Line 310: The p-values seem to be meaningless. Not sure they add any value here.

This is a good point and we have updated this analysis. Rather than exploring the probability that a feature changes the model we now explore how different features affect the model for the training, testing, and 'good' models. This change is detailed in response to a comment made by Reviewer #1.

The main changes are a new figure (Fig. 3, see response to Reviewer #3) and a new table (Table 3; shown in a response to Reviewer #1).

- Figure 8: Are these values volume-normalized? If not, they would skew toward surface values where grid boxes are smaller.

Yes, these are all volume weighted. This is now noted in the caption:

"Figure 9. Stacked, volume-weighted histograms showing the relative frequency distribution of dissolved [Ba] (A, B) and $\Omega$barite (C, D) in the global ocean."

- Section 5.1: It makes sense to remove models with lat and lon as predictors. After that, I disagree with all of the choices presented in this section, which ultimately lead to the choice of 1 model out of a possible 1,687 — talk about overfitting!

We agree with parts of this comment and disagree with others. We provide our reasoning in response to the next point.

- Eliminating models with Chl-a and MLD predictors: I will accept eliminating Chl-a, since including it degraded the median model. But just because including MLD only improved the average model by 3% is not a good reason to remove it as a predictor. You have a small sample size in the validation set, and MLD may encode key information for particular environments that are under-represented in the validation set. If it improves the model on average, it is reasonable to keep it.

This last point – *If it improves the model on average, it is reasonable to keep it* – got us thinking about the best way to approach the feature significance analysis, which is summarized in the new Table 3. We performed this analysis for the training data (random holdout cross folding), the testing data (regional cross validation), and for the 1,687 'good' models (also regional cross validation). We consider this last group the most relevant because:

"... these 1,687 models … are superior to existing methods for estimating [Ba] in seawater."

which we now state in Section 5.1. Looking at it this way, we see that six features improved the model on average ([PO$_4$], [NO$_3$], $T$, [O$_2$], $z$, [Si]), five degraded it (bathy., Chl. a, MLD, lat., and long.) , and one ($S$) had no effect. Since there is only one model that contains [PO$_4$], [NO$_3$], $T$, [O$_2$], $z$, and [Si], model #3112, we started with that. However, when we plotted the output from model #3112 it became clear that this model, while excellent (statistically speaking), was missing an important aspect of Ba geochemistry: input from rivers. We included some plots illustrating this comparison in the Supplement (see response to Reviewer #1) and note in Section 5.1.:

"Though volumetrically minor, riverine inputs are a geochemically important component of the marine Ba cycle, and the existence of nearshore Ba plumes underpins a major proxy application of Ba. Near-shore riverine influence is easily discerned by low S; we thus explored output from

model #3080, which is identical to model #3112, but includes S as a seventh feature during training. Models #3080 and #3112 exhibit identical statistical performance for the testing data (MAE = 4.3 nmol kg–1; Fig. S1) and make similar predictions for mean marine [Ba] and $\Omega$barite (89 nmol kg–1 and 0.82, respectively; see Supplement)."

- Eliminating models with Si eliminates the strongest predictor, which seems foolish. There is no reason to eliminate Si just because it appears in the definition of Ba*, which is not even in the target data. If you want the model to predict Ba* in addition to Ba, you could add that to the target when you train the models, but that is still no reason to remove Si from the predictor data (if it were, Si wouldn't even be in the list of features that you consider for this model).

Excellent point. We now retain Si as a feature when winnowing the list of good models.

As a result of this comment, we added [Si] to the model. The new model, #3080, is equivalent to model #3336+[Si], and the performance of the model is improved by about 3 %.

- The reason given for eliminating models with <=4 features is not valid. The analysis shows that *on average* the models with 5-8 predictors performed best (Figure 3). But that doesn't mean that there are not models with <5 predictors that could perform just as well and be just as probable (in fact there clearly are, as shown in Figure 3). It is arbitrary to eliminate these models.

This is a fair point and, given the changes made in response to an earlier comment ("*[i]f it improves the model on average, it is reasonable to keep it*") it no longer applies.

- In general, there is simply no good reason to choose 1 model as the "optimal" model. In fact the great benefit of the model testing that the authors have done is that it affords an ensemble of models from which to choose, many of them being equally or approximately equally probable. It one wants to "weight" the models one could do so be defining a probability function (MAD or something similar would do) and assigning a probability to each of the models. This would be better than simply choosing one single model (equivalent to assigning that model a probability of 1 and all the other models a probability of 0).

This is an interesting point. However, we did not implement this suggestion for two reasons. First, the analysis we performed in response to the reviewer's earlier comment showed that only six features consistently improved ML model performance—[PO$_4$], [NO$_3$], $T$, [O$_2$], $z$, [Si]. We thus decided to start with the only model that contained these six features (#3112). Adding $S$ to this model (#3080) had no effect on its MAE or MAPE, but it did mean that the model got rivers right. Second, while an ensemble of good models might be interesting, we believe that end users of this data product may find it easier to simply use our 'best estimate' of marine [Ba]. As more data become available, such as from the next GEOTRACES IDP release, a new best model could well emerge and we can update the model output incorporating those data.

- Line 414: Figure 3 doesn't show sea surface Ba.

Good catch.

This particular cross reference has been cut.

- Line 426: Or maybe the model is just wrong in those regions. Do any other of the possible models (e.g., not model #3336) show elevated Ba at those locations?

Great point.

We now have a section in the Supplement showing a comparison of ML model outputs close to the mouths of major rivers. It appears that including $S$ is important if the models are to recognize that there is elevated [Ba] close to shore.

- Line 430: Sure, it's reasonable. It's just unreasonable to say that there are no other possibilities.

We've added another possibility to this section (underlined text):

"The reasons for the lack of elevated [Ba] near the outflow of these two rivers is less clear. It is possible that the model is simply inaccurate in these regions, though we have no particular reason to believe that this is the case. Alternatively, it may reflect seasonal variations in Ba release that are not captured by our mean annual model (e.g., Joung & Shiller, 2014). It could also indicate that these particular rivers are not major net sources of Ba to the surface ocean, which might be the case if dissolved Ba is being retained in the catchment (e.g., Charbonnier et al., 2020) or estuary (e.g., Coffey et al., 1997)."

- Line 551: It would be better to base such uncertainties on an ensemble of most-probable models (rather than a single model)

Based on the revised feature significance analysis suggested by the reviewer, we restricted our analysis to a single model (#3080) and base our uncertainties on the generalization error.

No changes made.

Citations added.